# METIS: TRAINING LLMS WITH FP4 QUANTIZATION

**Hengjie Cao**[1,3,*]    **Mengyi Chen**[1,*]    **Yifeng Yang**[1,*]    **Ruijun Huang**[1]    **Fang Dong**[1]
**Jixian Zhou**[1]    **Anrui Chen**[1]    **Mingzhi Dong**[2]    **Yujiang Wang**[4,5]    **Jinlong Hou**[3]
**Yuan Cheng**[3]    **Fan Wu**[6]    **Fan Yang**[1]    **Tun Lu**[1]    **Ning Gu**[1]    **Li Shang**[1,7,†]

[1] Fudan University    [2] University of Bath    [3] Shanghai Innovation Institute
[4] University of Oxford    [5] Oxford Suzhou Centre for Advanced Research
[6] Huawei    [7] Shenzhen Loop Area Institute

## ABSTRACT

This work identifies anisotropy in the singular value spectra of parameters, activations, and gradients as the fundamental barrier to low-bit training of large language models (LLMs). These spectra are dominated by a small fraction of large singular values, inducing wide numerical ranges that cause quantization bias and severe spectral distortion, ultimately degrading training performance. This work presents *Metis*, a spectral-domain quantization framework that partitions anisotropic spectra into narrower sub-distributions for independent quantization, thereby reducing errors and preserving spectral structure. To minimize overhead, Metis leverages two key properties of the dominant spectral subspace: preservation via sparsely random sampling and preservation via random projection, reducing decomposition cost to a negligible level. On LLaMA-3 8B trained with 100B tokens, Metis enables robust W4A4G4 training with FP4 quantization of weights, activations, and gradients, yielding only a 0.4% training loss gap and a 0.1% degradation in downstream accuracy relative to BF16. Beyond matching BF16 fidelity, Metis also surpasses Nvidia's FP4 recipe, consistently achieving lower loss and higher downstream accuracy while incurring significantly lower computational overhead. The code implementation for Metis is available at: https://github.com/sii-research/Metis.

## 1 INTRODUCTION

Training large language models (LLMs) with low-bit quantization of parameters, activations, and gradients offers substantial gains in efficiency, cost, and scalability. In recent years, progress has advanced from FP32 to BF16 and, more recently, to FP8 training (Micikevicius et al., 2022; Peng et al., 2023; Perez et al., 2023). Looking ahead, Nvidia's Blackwell technical report shows that the NVFP4 format reduces memory consumption by 1.8× and accelerates General Matrix Multiplications (GeMM) by 7× compared to FP8, underscoring the efficiency potential of FP4 training (Alvarez et al.; Devleker & Ghodsian). However, pushing the training frontier further down to FP4 is not a straightforward continuation: FP4 imposes exponentially tighter constraints on precision and dynamic range, which conflict with the inherently wide distributions of parameters, activations, and gradients.

This study investigates the origins of these wide distributions and analyzes their impact on training stability and effectiveness under FP4 quantization. The key findings are outlined below and visualized in Fig. 1.

**Anisotropy is universal in modern LLMs.** In weight, activation, and gradient matrices, a small fraction of singular values dominate, yielding a highly imbalanced spectrum. This phenomenon is consistently observed across model architectures and parameter scales up to 671B.

**Anisotropy induces wide numerical distributions.** Wide distributions of weights, activations, and gradients arise from the superposition of singular components: larger components contribute to the

---

*Equal contribution.

†Corresponding author. Email: lishang@fudan.edu.cn.

large-value region, whereas smaller ones concentrate near zero. This spread originates from variability in singular values, which projects aligned components into entries of corresponding magnitudes.

**Quantization bias induces spectral distortion.** Commonly used block-level low-bit quantization introduces bias that disproportionately favors large values, thereby reducing the effective resolution for small values. Under anisotropy, this effect leads to severe distortion in spectral space: smaller singular components incur substantially larger value errors and direction perturbations than their larger counterparts.

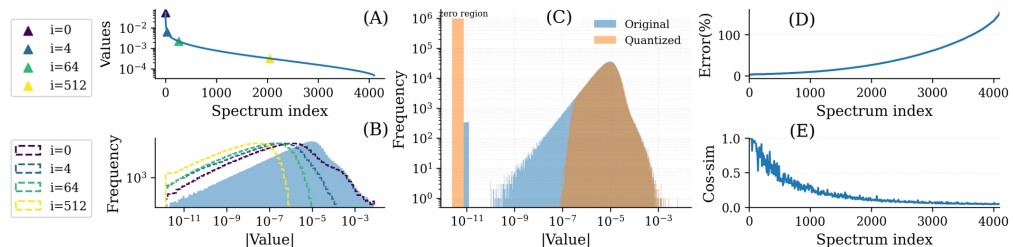

Figure 1: Overview of anisotropy and its impact on quantization, illustrated using a gradient matrix from LLaMA-3 8B. (A) Singular value spectrum exhibits strong anisotropy, with a few singular values dominating the spectrum. (B) The wide matrix distribution arises from the superposition of singular components: large components (e.g., i=0) drive the high-value region, while small components concentrate near zero. (C) Quantization bias disproportionately rounds many small values to zero. (D–E) In spectral space, smaller singular components incur substantially larger relative quantization errors in singular values and more severe perturbations in singular directions. Details corresponding to (A) in Section 2.1, (B) in Section 2.2, and (C–E) in Section 2.3.

Inspired by these findings, we propose *Metis*, a quantization framework that preserves the spectral structure under FP4 formats and allows robust FP4 training. Metis operates in the spectral domain, applying decomposition to weight, activation, and gradient matrices to disentangle dominant from long-tailed singular components, thereby allowing quantization over substantially narrower distributions. A central challenge is the computational complexity of spectral decomposition. To address this, we leverage two key structural properties revealed in our empirical study:

(i) *Subspace Preservation via Sparsely Random Sampling*, where the dominant subspace estimated from a sparsely random sampled subset is reliably generalized to the whole batch;

(ii) *Subspace Preservation via Random Projection*, where the dominant subspace can be faithfully captured within a reduced hidden dimension via random projection.

Building on these insights, Metis renders the decomposition cost negligible by employing sparse random sampling over sequences and random projections on the hidden dimension, each reducing complexity by approximately two orders of magnitude.

Metis enables robust W4A4G4 training by quantizing all GeMM matrices to 4-bit floating point. On an 8B LLaMA-3 model (et al., 2024) trained with 100B tokens from the DCLM dataset (Li et al., 2025), Metis narrows the gap to BF16 to only a 0.4% increase in training loss and a 0.1% degradation in downstream accuracy. Compared to our implementation of NVIDIA's FP4 recipe (Devleker & Ghodsian), Metis consistently achieves lower training loss and higher downstream accuracy while incurring significantly less computational overhead.

## 2 ANALYSIS

Anisotropy emerges as a key structural factor underlying the wide distributions observed in weights, activations, and gradients. This section analyzes these anisotropic matrices and examines how they misalign with low-bit quantization schemes. Unless otherwise specified, all experiments are conducted on the LLaMA-3 8B model trained on 100B tokens from the DCLM dataset.

## 2.1 ANISOTROPY: A UNIVERSAL PROPERTY OF MODERN LLMS

For a matrix $\mathbf{M} \in \mathbb{R}^{m \times n}$, we perform Singular Value Decomposition (SVD) to obtain singular values $\{\sigma_i\}_{i=1}^{\min(m,n)}$, left singular vectors $\{\mathbf{u}_i\} \in \mathbb{R}^m$, and right singular vectors $\{\mathbf{v}_i\} \in \mathbb{R}^n$, such that $\mathbf{M} = \sum_{i=1}^{\min(m,n)} \sigma_i \mathbf{u}_i \mathbf{v}_i^\top$. We assume singular values are sorted in descending order, i.e., $\sigma_1 \geq \sigma_2 \geq \cdots \geq \sigma_r > 0$ with $r = \min(m, n)$.

Anisotropy is characterized by a spectrum where a few leading singular values dominate, yielding a highly imbalanced distribution across directions. Previous studies have reported anisotropy in activation matrices (Ethayarajh, 2019; Mu et al., 2017; Puccetti et al., 2022; Rudman & Eickhoff, 2023; Yu et al., 2021), and our analysis further demonstrates that it is universal across weights, activations, and gradients. As shown in Figure 2 (A), their singular value spectra exhibit pronounced anisotropy, with only 0.63%, 3.15%, and 2.91% of components (identified by the elbow point of maximum curvature) dominating the spectra of weights, activations, and gradients, respectively. We further validate this pattern on publicly released models, including the Qwen family (Bai et al., 2023) and DeepSeek-R1 (Liu et al., 2024), where weight matrices from 7B to 671B parameters consistently show fewer than 3% of singular values dominating the spectrum, confirming anisotropy as a universal property across architectures and scales.

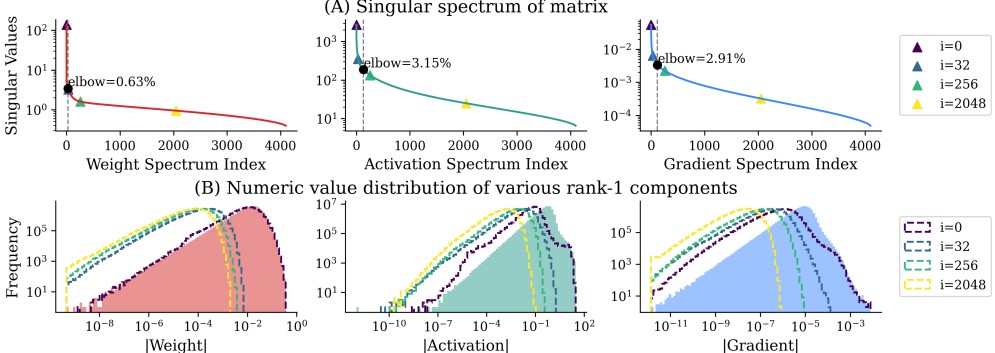

Figure 2: Analysis of weight, activation, and gradient matrices (layer 32, FeedForward(FFN)). (A) The singular value spectra exhibit strong anisotropy, with only 0.63%, 3.15%, and 2.91% of components (identified by the elbow point of maximum curvature) dominating the spectrum. (B) Filled regions denote full-matrix distributions; dashed histograms showes selected rank-1 components $(\mathbf{u}_i \sigma_i \mathbf{v}_i^\top$ for $i = 0, 16, 128, 1024)$. Dominant components (e.g., $i = 0$) drive the high-value region, while smaller ones (e.g., $i = 1024$) contribute near zero. See A.1 for additional results.

## 2.2 ANISOTROPY INDUCES WIDE NUMERICAL DISTRIBUTIONS

The wide distributions of weights, activations, and gradients arise from the superposition of singular components $(\mathbf{u}_i \sigma_i \mathbf{v}_i^\top)$: dominant components (e.g., $i = 0$) account for the large-value region, whereas smaller components (e.g., $i = 1024$) concentrate near zero, as illustrated in Fig. 2 (B).

**Wide distributions arise from the skewed singular value spectrum.** This spread originates from variability in singular values, which project aligned components into entries of corresponding magnitudes. In the SVD decomposition $\mathbf{M} = \sum_i \sigma_i \mathbf{u}_i \mathbf{v}_i^\top$, large singular values amplify aligned components into high-magnitude entries, whereas small singular values suppress others toward near zero, thereby producing long-tailed distributions. After isolating the impact of singular values, Fig. 3(A) shows that the singular vectors all exhibit similar shapes with widths far narrower than that of the full matrix. Thus, the wide ranges are a direct consequence of the skewed singular spectrum.

**Residual singular components confirm the dominant subspace effect.** To validate this mechanism, we analyze residual matrices, corresponding to the long-tail singular components, after subtracting the leading singular components amplified by large singular values. As shown in Fig. 3 (B), removing the top 3% of components, which dominate the spectrum, yields residual distributions one to two orders of magnitude narrower than those of the full matrix. This demonstrates that wide ranges arise primarily from the dominant subspace, while the residual remains quantization-friendly.

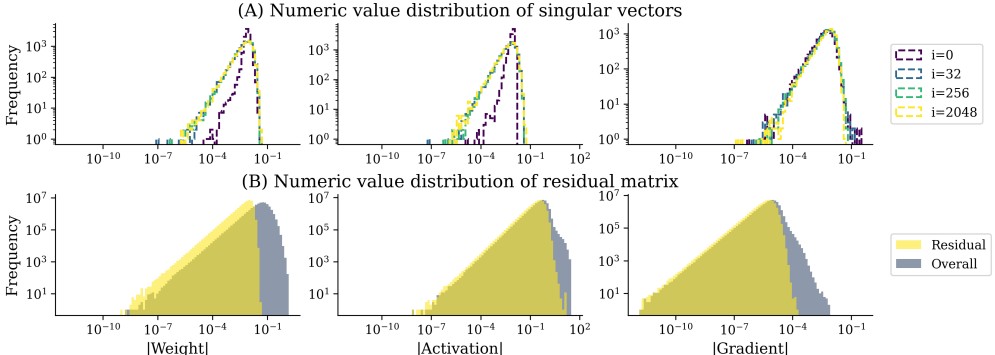

Figure 3: Analysis of weight, activation, and gradient matrices with hidden dimension 4096 (layer 32, FFN). (A) Left singular vector distributions: all exhibit similar shapes with widths much smaller than that of the full matrix. (B) Yellow regions show the residuals after removing the top 128 components ($3\% \times 4096 \approx 123$, rounded to the nearest power of two), while the grey region represents the original matrix distribution. The residuals are one to two orders of magnitude narrower than the full matrix, confirming that wide ranges originate from dominant components. More results in A.2 .

## 2.3 QUANTIZATION BIAS: SPECTRAL DISTORTION

In extreme low-bit regimes such as FP4, block-wise quantization is a commonly-used approach. By restricting scaling to small blocks with fewer entries, the likelihood of encountering extreme values is reduced, resulting in narrower local distributions and finer-grained scaling that mitigates quantization error (Rouhani et al., 2023; Nvidia, 2025). Moreover, fixed-size blocks align naturally with Tensor Core execution, enabling efficient parallelism (Rouhani et al., 2023; Nvidia, 2025).

Formally, a $b$-bit quantizer $\mathcal{Q}_b$ maps a matrix $\mathbf{M} \in \mathbb{R}^{m \times n}$ to its quantized form $\overline{\mathbf{M}} = \mathcal{Q}_b(\mathbf{M})$. Throughout the paper, we will use the bar symbol to denote the matrix after quantization for simplicity. The matrix is partitioned into fixed-size blocks (e.g., size 16 in the NVFP4 format (Nvidia, 2025)). For each block $\mathbf{E} \in \mathbb{R}^t$, NVFP4 selects a single scaling factor $s$ in FP8-E4M3, set by the block's maximum magnitude and rounded up to the nearest representable FP8 value to ensure coverage. Each element is then quantized as $\mathcal{Q}_4^{\text{NV}}(\mathbf{e}_i) = \text{round}\left(\frac{\mathbf{e}_i}{s}\right) \cdot s$, where $\text{round}(\cdot)$ denotes rounding each scaled element to the nearest FP4-E2M1 representable value.

**Quantization bias.** In broad distributions, where the range can span multiple orders of magnitude, as observed in weight, activation, and gradient matrices in Fig. 2, determining the scaling factor $s$ by the block maximum introduces bias. This bias disproportionately favors large values while suppressing the resolution available for small ones. As shown in Fig. 1 (C), nearly half of the values are rounded entirely to zero after quantization, resulting in the destructive loss of the information they represent.

**Spectral distortion.** In an anisotropic matrix with uneven singular value distribution, quantization bias causes severe distortion in spectral space, especially for small singular components. As shown in Fig. 1 (D) (E), all components suffer relative errors in singular values and directional perturbations, with smaller ones exhibiting greater distortion in both magnitude and direction.

## 3 METIS

We propose *Metis*, an FP4 training framework designed to address the challenge of spectral anisotropy. Metis partitions the spectrum into narrower sub-distributions and applies quantization independently within each. This design yields two benefits: (i) as shown in Fig. 3, each sub-distribution is significantly narrower than the full spectrum, reducing quantization error; and (ii) it prevents small singular components from being overwhelmed by large ones, thereby mitigating perturbations in singular values and preserving directional consistency within the subspace.

The central challenge, however, lies in the high cost of repeatedly performing spectral decomposition during training. Section 3.1 shows that anisotropy induces two key properties of the dominant subspace, preservation via sparsely random sampling and random projection, making scalable

spectral decomposition feasible. Section 3.2 integrates this scalable decomposition into GeMM, the source of over 95% of the training workload in large language models (Vaswani et al., 2017; Wang et al., 2025), applying FP4 quantization to all GeMMs in both forward and backward passes. Section 3.3 analyzes the additional computational complexity introduced by Metis.

### 3.1 ENABLING PROPERTIES FOR SCALABLE SPECTRAL DECOMPOSITION

The prohibitive cost of spectral decomposition presents a major obstacle to deploying spectral-domain quantization at scale. Let $\mathbf{X} \in \mathbb{R}^{l \times m}$ denote the activation tensor, where $l = b \cdot s$ with $b$ the batch size, $s$ the sequence length, and $m$ the hidden dimension. In practice, $l$ can reach millions while $m$ is typically in the thousands. Performing a full SVD on such matrices at every iteration incurs a cost of $\mathcal{O}(lm^2)$, making direct application impractical.

However, anisotropy offers a crucial opportunity: it causes a small number of singular values $k$ to dominate, with $k \ll l, m$, so spectral decomposition only needs to isolate the corresponding dominant subspace, yielding markedly narrower spectral sub-distributions. Building on this, our further investigation reveals two properties that enable dimension reduction along both the sequence and hidden axes: (i) the dominant subspace estimated from a sparsely sampled subset generalizes reliably to the full batch, and (ii) the dominant subspace can be faithfully captured within a reduced hidden dimension via random projection.

**Subspace Preservation via Sparsely Random Sampling.** The sequence dimension $l$ is typically orders of magnitude larger than the dominant subspace dimension $k$. Intuitively, this abundance of samples ensures that the dominant subspace can be estimated stably even from a small random subset of samples. From a theoretical perspective, the anisotropic structure guarantees that random subsampling preserves the dominant subspace with high probability. Let $\Sigma = \frac{1}{l}\mathbf{X}^\top\mathbf{X} \in \mathbb{R}^{m \times m}$ be the covariance. For a random subset $\mathbf{X}_\Omega \in \mathbb{R}^{l_k \times n}$ with $l_k \ll l$, the sample covariance $\hat{\Sigma} = \frac{1}{l_k}\mathbf{X}_\Omega^\top\mathbf{X}_\Omega$ satisfies the matrix Chernoff (Tropp, 2012):

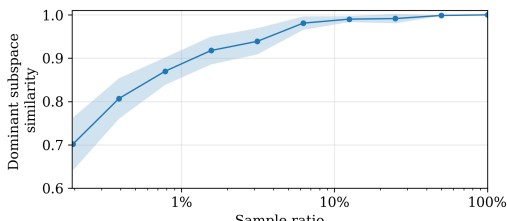

Figure 4: Subspace alignment between the dominant subspace of the full batch and that of randomly sampled subsets of sequences. Alignment quickly saturates as the sample ratio increases, with just 1% of sequences achieving nearly 0.9 alignment with the full-batch subspace. More results in B.1.

$$\Pr\left[\|\hat{\Sigma} - \Sigma\|_2 \leq \epsilon\|\Sigma\|_2\right] \geq 1 - \delta, \text{ whenever } l_k = \mathcal{O}\left(\frac{n}{\epsilon^2}\log\frac{n}{\delta}\right).$$

Combined with the Davis–Kahan theorem (Davis & Kahan, 1970), $\|\mathbf{A}_k\mathbf{A}_k^\top - \hat{\mathbf{A}}_k\hat{\mathbf{A}}_k^\top\|_2 \leq \frac{\|\hat{\Sigma}-\Sigma\|_2}{\Delta}$, where $\mathbf{A}_k$ and $\hat{\mathbf{A}}_k \in \mathbb{R}^{m \times k}$ are the top-$k$ eigenspaces of $\Sigma$ and $\hat{\Sigma}$, and $\Delta$ is the eigengap, this guarantees that the dominant subspace estimated from a small random subset faithfully recovers that of the full batch. Empirically, we measure the alignment of the dominant subspace between the full batch and randomly sampled subsets using the mean squared canonical correlation between their orthonormal bases. We observe rapid saturation as the sample ratio increases, where the sample ratio is defined as the fraction of tokens included in the sampled subset relative to the total number of tokens in the global batch. For instance, as shown in Fig. 4, sampling only 1% of sequences achieves nearly 0.9 overlap with the full-batch subspace, demonstrating the effectiveness of this subspace approximation.

**Subspace Preservation via Random Projection.** We further establish that the dominant spectral structure remains stable under random projections of the hidden dimension. Let $\Omega \in \mathbb{R}^{n \times (k+s)}$ be a Gaussian test matrix with oversampling parameter $s$, and form the sketch $\mathbf{Z} = \mathbf{X}\Omega$ with thin QR factorization $\mathbf{Z} = \mathbf{H}\mathbf{R}$. Randomized SVD theory (Halko et al., 2011) shows that, with high probability, the projection error $\|(\mathbf{I} - \mathbf{H}\mathbf{H}^\top)\mathbf{X}\|_2$ is controlled by the tail singular values of $\mathbf{X}$. Furthermore, by Davis–Kahan, the subspace error for the top-$k$ eigenspace satisfies $\|\mathbf{A}_k\mathbf{A}_k^\top - (\mathbf{H}\tilde{\mathbf{A}}_k)(\mathbf{H}\tilde{\mathbf{A}}_k)^\top\|_2 \leq \frac{2\|(\mathbf{I}-\mathbf{H}\mathbf{H}^\top)\mathbf{X}\|_2}{\Delta}$, where $\tilde{\mathbf{A}}_k \in \mathbb{R}^{m \times k}$ are the top-$k$ eigenspaces of $\mathbf{H}^\top\Sigma\mathbf{H}$. This demonstrates that a modest oversampling parameter $s$ and a nontrivial spectral gap suffice to ensure that Gaussian projections preserve the dominant subspace with high fidelity, enabling accurate recovery without operating in the full ambient dimension $m$.

**Scalable Spectral Decomposition.** Leveraging these properties, we adopt a two-step procedure for efficient decomposition. First, *sparse random sampling* selects less than 1% of sequences to estimate the dominant subspace, which is then broadcast to the full batch. Second, we apply *randomized SVD* to recover the top-$k$ components. This reduces the complexity from $\mathcal{O}(ln^2)$ to $\mathcal{O}(l_k nk)$, while the additional cost remains asymptotically negligible compared with forward and backward GeMM.

**Temporal Reuse Across Steps.** We further observe strong temporal stability in the dominant spectral structure: the leading low-rank subspace of both activation and output-gradient matrices remains highly aligned across adjacent steps. Thus, after one spectral decomposition, we reuse the extracted low-rank factors for several subsequent steps. In practice, we refresh the decomposition every 8 steps, further reducing the amortized overhead.

## 3.2 Spectral Decomposition

Each matrix in GeMM is decomposed into a low-rank component and a residual: low-rank singular values are kept in high precision, while the associated singular vectors and residual are quantized to low bit. Metis handles parameters, activations, and gradients differently: for parameters, low-rank and residual parts are stored as separate trainable variables and updated independently; for activations and gradients, spectral decompositions are recomputed dynamically at each iteration.

**Forward pass.** Let $\mathbf{W} \in \mathbb{R}^{m \times n}$ denote a weight matrix and $\mathcal{X} \in \mathbb{R}^{b \times s \times m}$ an input activation tensor, where $b$ is the batch size, $s$ the sequence length, and $m$ the hidden dimension. For the GeMM operation, $\mathcal{X}$ is reshaped into a matrix $\mathbf{X} \in \mathbb{R}^{l \times m}$ with $l = bs$. The output is then computed as $\mathbf{Y} = \mathbf{X}\mathbf{W} \in \mathbb{R}^{l \times n}$. Our goal is to quantize this GeMM computation under low-bit formats.

A rank-$k$ approximation of $\mathbf{W}$ is given by $\hat{\mathbf{W}}_k = \mathbf{U}_k \mathbf{S}_k \mathbf{V}_k^\top$, where $\mathbf{U}_k \in \mathbb{R}^{m \times k}$, $\mathbf{V}_k \in \mathbb{R}^{n \times k}$, and $\mathbf{S}_k = \text{diag}(\sigma_1, \ldots, \sigma_k) \in \mathbb{R}^{k \times k}$. The residual is $\mathbf{W}_\text{R} = \mathbf{W} - \hat{\mathbf{W}}_k$, so that $\mathbf{W} = \hat{\mathbf{W}}_k + \mathbf{W}_\text{R} = \mathbf{U}_k \mathbf{S}_k \mathbf{V}_k^\top + \mathbf{W}_\text{R}$. Similarly, the activation matrix $\mathbf{X}$ is decomposed as: $\mathbf{X} = \hat{\mathbf{X}}_k + \mathbf{X}_\text{R} = \mathbf{A}_k \mathbf{\Lambda}_k \mathbf{B}_k^\top + \mathbf{X}_\text{R}$, where $\mathbf{A}_k \in \mathbb{R}^{l \times k}$, $\mathbf{B}_k \in \mathbb{R}^{m \times k}$ and $\mathbf{\Lambda}_k \in \mathbb{R}^{k \times k}$.

Accordingly, the forward GeMM can be written as

$$\mathbf{Y} = (\mathbf{A}_k \mathbf{\Lambda}_k \mathbf{B}_k^\top + \mathbf{X}_\text{R})(\mathbf{U}_k \mathbf{S}_k \mathbf{V}_k^\top + \mathbf{W}_\text{R}) \tag{1}$$

The quantization function $\mathcal{Q}_b$ is then applied separately to matrices in Eq. 1 other than $\mathbf{S}_k$ and $\mathbf{\Lambda}_k$. Specifically, the forward computation of $\mathbf{Y}$ under $b$-bit quantization, denoted as $\hat{\mathbf{Y}}$, is computed as

$$\begin{aligned}
\hat{\mathbf{Y}} &= (\mathcal{Q}_b(\mathbf{A}_k)\mathbf{\Lambda}_k \mathcal{Q}_b(\mathbf{B}_k^\top) + \mathcal{Q}_b(\mathbf{X}_\text{R}))(\mathcal{Q}_b(\mathbf{U}_k)\mathbf{S}_k \mathcal{Q}_b(\mathbf{V}_k^\top) + \mathcal{Q}_b(\mathbf{W}_\text{R})) \\
&= \overline{\mathbf{X}}\overline{\mathbf{U}}_k \mathbf{S}_k \overline{\mathbf{V}}_k^\top + \overline{\mathbf{X}}\overline{\mathbf{W}}_\text{R}.
\end{aligned} \tag{2}$$

**Backward pass.** In backward propagation, we quantize the GeMM operations associated with derivative computations of matrices in Eq. 1. Formally, denote the derivatives of the loss function $\mathcal{L}$ with respect to (w.r.t) $\mathbf{Y}$ as $\mathbf{D} = \frac{\partial \mathcal{L}}{\partial \mathbf{Y}} \in \mathbb{R}^{l \times n}$, we need to compute the following derivatives for updating parameters: $\frac{\partial \mathcal{L}}{\partial \mathbf{X}}$, $\frac{\partial \mathcal{L}}{\partial \mathbf{U}_k}$, $\frac{\partial \mathcal{L}}{\partial \mathbf{S}_k}$, $\frac{\partial \mathcal{L}}{\partial \mathbf{V}_k}$, and $\frac{\partial \mathcal{L}}{\partial \mathbf{W}_\text{R}}$. Similarly, we first decompose $\mathbf{D}$ into a SVD low-rank approximation and the residual, which is $\mathbf{D} = \mathbf{P}_k \mathbf{T}_k \mathbf{Q}_k^\top + \mathbf{D}_\text{R}$, where $\mathbf{P}_k \in \mathbb{R}^{l \times k}$, $\mathbf{Q}_k \in \mathbb{R}^{n \times k}$, $\mathbf{T}_k \in \mathbb{R}^{k \times k}$. The derivative $\frac{\partial \mathcal{L}}{\partial \mathbf{X}}$ is computed as

$$\begin{aligned}
\frac{\partial \mathcal{L}}{\partial \mathbf{X}} &= \frac{\partial \mathcal{L}}{\partial \mathbf{Y}} \frac{\partial \mathbf{Y}}{\partial \mathbf{X}} = \mathbf{D}(\mathbf{V}_k \mathbf{S}_k^\top \mathbf{U}_k^\top + \mathbf{W}_\text{R}^\top), \\
&= \mathbf{P}_k \mathbf{T}_k \mathbf{Q}_k^\top \mathbf{V}_k \mathbf{S}_k^\top \mathbf{U}_k^\top + \mathbf{P}_k \mathbf{T}_k \mathbf{Q}_k^\top \mathbf{W}_\text{R}^\top + \mathbf{D}_\text{R} \mathbf{V}_k \mathbf{S}_k^\top \mathbf{U}_k^\top + \mathbf{D}_\text{R} \mathbf{W}_\text{R}^\top.
\end{aligned}$$

We then compute $\frac{\partial \mathcal{L}}{\partial \mathbf{X}}$ under 4-bit quantization as

$$\frac{\hat{\partial \mathcal{L}}}{\partial \mathbf{X}} = \overline{\mathbf{P}}_k \mathbf{T}_k \overline{\mathbf{Q}}_k^\top \overline{\mathbf{V}}_k \mathbf{S}_k^\top \overline{\mathbf{U}}_k^\top + \overline{\mathbf{P}}_k \mathbf{T}_k \overline{\mathbf{Q}}_k^\top \overline{\mathbf{W}}_\text{R}^\top + \overline{\mathbf{D}}_\text{R} \overline{\mathbf{V}}_k \mathbf{S}_k^\top \overline{\mathbf{U}}_k^\top + \overline{\mathbf{D}}_\text{R} \overline{\mathbf{W}}_\text{R}^\top. \tag{3}$$

Following a similar pathway, we compute other derivatives under 4-bit quantization as follows:

$$\frac{\hat{\partial \mathcal{L}}}{\partial \mathbf{U}_k} = \overline{\mathbf{X}}^\top \overline{\mathbf{P}}_k \mathbf{T}_k \overline{\mathbf{Q}}_k^\top \overline{\mathbf{V}}_k \mathbf{S}_k^\top + \overline{\mathbf{X}}^\top \overline{\mathbf{D}}_\text{R} \overline{\mathbf{V}}_k \mathbf{S}_k^\top, \quad \frac{\hat{\partial \mathcal{L}}}{\partial \mathbf{S}_k} = \overline{\mathbf{U}}_k^\top \overline{\mathbf{X}}^\top \overline{\mathbf{P}}_k \mathbf{T}_k \overline{\mathbf{Q}}_k^\top \overline{\mathbf{V}}_k + \overline{\mathbf{U}}_k^\top \overline{\mathbf{X}}^\top \overline{\mathbf{D}}_\text{R} \overline{\mathbf{V}}_k,$$

$$\frac{\hat{\partial \mathcal{L}}}{\partial \mathbf{V}_k} = \mathbf{S}_k^\top \overline{\mathbf{U}}_k^\top \overline{\mathbf{X}}^\top \overline{\mathbf{P}}_k \mathbf{T}_k \overline{\mathbf{Q}}_k^\top + \mathbf{S}_k^\top \overline{\mathbf{U}}_k^\top \overline{\mathbf{X}}^\top \overline{\mathbf{D}}_\text{R}, \quad \frac{\hat{\partial \mathcal{L}}}{\partial \mathbf{W}_\text{R}} = \overline{\mathbf{X}}^\top \overline{\mathbf{P}}_k \mathbf{T}_k \overline{\mathbf{Q}}_k^\top + \overline{\mathbf{X}}^\top \overline{\mathbf{D}}_\text{R}.$$

## 3.3 DISCUSSION ON TRAINING EFFICIENCY

We analyze Metis's additional computational overhead, arising from (i) the extra small-scale GeMMs introduced by the decomposition and (ii) the decomposition itself. To mitigate this cost, we further develop a customized optimized operator for low-rank spectral decomposition; detailed optimization strategies are described in B.2. Kernel-level overhead comparisons and end-to-end empirical results are reported in B.3.

**Forward pass.** In the baseline, evaluating $\mathbf{Y} = \mathbf{XW}$ requires $\mathcal{O}(lmn)$ operations. Under Metis, the forward computation follows Eq. 1, where the additional mixed products introduce $\mathcal{O}(lmk + mnk + lnk)$ overhead on top of the baseline cost. Moreover, the activation decomposition performed at each step adds $\mathcal{O}(l_k mk)$, where $l_k \ll l$ by sparse random sampling.

**Backward pass.** In the baseline, gradients with respect to $\mathbf{X}$ and $\mathbf{W}$ require $\mathcal{O}(lmn)$ operations. Under Metis, gradient computation follows Eq. 3, where mixed products contribute $\mathcal{O}(lmk + mnk + lnk)$. In addition, output gradient decomposition at each step adds $\mathcal{O}(l_k nk)$.

**Overall complexity.** Combining forward and backward contributions, Metis introduces an additional cost of $\mathcal{O}(lmk + mnk + lnk + l_k mk + l_k nk)$ per training step, which is asymptotically much smaller than the baseline $\mathcal{O}(lmn)$, making Metis tractable at the scale of modern LLMs.

## 4 EXPERIMENTS

Experimental results show that Metis under FP4 training maintains, and in some cases improves, both training loss and downstream accuracy relative to BF16 baselines, supporting FP4 as a viable setting for large-scale model training.

**Models and Datasets.** We conduct experiments on GPT-2 (130M and 1.1B) (Radford et al., 2019) and LLaMA-3 (8B) (et al., 2024). For pretraining, we use the DCLM (Li et al., 2025) dataset and train each model on 100B tokens. The raw data are segmented to split long documents and concatenate short ones, filtered to remove non-Unicode and non-English content, and further processed with Qwen3 to discard entries whose perplexity deviates by more than two standard deviations from the mean. For downstream evaluation, we consider three task types: question answering, classification, and cloze prediction. For question answering, we use ARC (Clark et al., 2018), RACE (Lai et al., 2017), and BoolQ (Clark et al., 2019); for classification, we use HellaSwag (Zellers et al., 2019) and PIQA (Bisk et al., 2019); and for cloze prediction, we use LAMBADA (OpenAI) (Kazemi et al., 2023).

**FP4 Training.** The efficacy of Metis under FP4 quantization is evaluated against both FP4 and BF16 baselines. All FP4 training in this work adopts W4A4G4 quantization, where weights, activations, and gradients are represented in the E2M1 NVFP4 format. Our main experiments with NVFP4 are conducted through simulation in BF16. Stochastic rounding (SR) is applied by default in all FP4 experiments, as it mitigates quantization bias, is orthogonal to other meth-

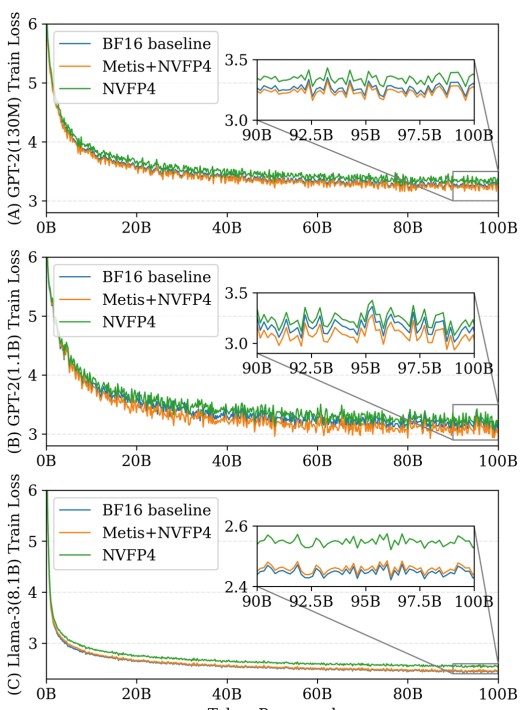

Figure 5: Training loss curves for (A) GPT-2 130M, (B) GPT-2 1.1B, and (C) LLaMA-3 8B. Direct NVFP4 incurs a loss gap of 3–4% relative to BF16 baseline, while Metis reduces the gap to 0.4% on LLaMA-3 and even slightly surpasses the BF16 baseline on GPT-2 models.

ods. The Metis rank is fixed at 1.5% for low-rank approximation in both forward and backward passes, as our sensitivity analysis in C.2 shows that 1.5% is sufficient to maintain performance.

## 4.1 Main results

**Training Loss.** Fig. 5 shows the training loss curves of the BF16 baseline, NVFP4, and NVFP4 + Metis, while Table 1 reports the corresponding final test losses. NVFP4 exhibits a clear gap relative to the BF16 baseline, particularly on LLaMA-3 8B, with test loss increasing by 3–4% across all models. In contrast, Metis narrows the gap to 0.4% on LLaMA-3 8B and even achieves slightly lower loss than the BF16 baseline on GPT-2 models. This expressivity enhancement of Metis over the BF16 baseline may stem from the separation of low-rank and residual branches of weight matrices during training, which reduces interference between feature subspaces and contributes to the observed performance gains. We further examine the spectrum of the residual matrix to test whether anisotropy re-emerges. The results in C.3 show it does not: the singular value spectrum of the residual matrix remains flat, indicating that anisotropy is effectively addressed by the low-rank branch.

**Performance on Downstream Tasks.** As shown in Table 1, direct NVFP4 quantization results in an average drop of 1% relative to the BF16 baselines across all models. In contrast, Metis consistently outperforms these FP4 baselines, reducing the drop to 0.1% on LLaMA-3 8B and even slightly surpassing the BF16 baseline on GPT-2 models. These findings are consistent with the loss results reported earlier and demonstrate Metis's ability not only to preserve but, in some cases, to enhance model expressiveness under ultra-low-bit constraints.

Table 1: Downstream performance across different settings, reported with task-specific metrics. Direct NVFP4 quantization leads to an average drop of 1% relative to the BF16 baseline, while Metis reduces the gap to 0.1% on LLaMA-3 and slightly surpasses BF16 on GPT-2 models.

| Model | Method | Loss | ARC-C | ARC-E | BoolQ | LAMBADA | PIQA | RACE | HellaSwag | Avg |
|---|---|---|---|---|---|---|---|---|---|---|
| GPT-2 | BF16 | 3.23 | **24.3** | **32.1** | 60.4 | 31.3 | **62.1** | 47.3 | **32.5** | 41.4 |
| (130M) | FP4 | 3.32 | 22.9 | 31.5 | **60.7** | 31.2 | 60.9 | 47.0 | 31.7 | 40.8 |
| | FP4-Metis | **3.20** | 24.1 | 31.9 | 60.2 | **31.8** | 61.8 | **48.4** | 32.2 | **41.5** |
| GPT-2 | BF16 | 3.09 | **30.7** | 59.5 | **60.0** | 35.4 | **64.9** | 47.3 | 41.1 | 48.4 |
| (1.1B) | FP4 | 3.22 | 29.8 | 57.6 | 59.8 | 35.5 | 63.6 | 46.8 | 39.5 | 47.5 |
| | FP4-Metis | **3.01** | 30.1 | **60.0** | 59.5 | **36.2** | 64.1 | **48.5** | 41.9 | **48.6** |
| LlaMa-3 | BF16 | **2.44** | 32.4 | **60.4** | 59.2 | **37.6** | 70.5 | **47.0** | **50.9** | **51.1** |
| (8B) | FP4 | 2.53 | 31.3 | 58.5 | 58.2 | 36.5 | 69.8 | 46.9 | 49.4 | 50.0 |
| | FP4-Metis | 2.45 | **32.7** | 59.6 | **59.8** | 37.0 | **70.9** | 46.5 | 50.7 | 51.0 |

## 4.2 Ablation Study

We ablate the two key components of Metis: Spectral Decomposition and Sparse Random Sampling.

**Spectral Decomposition.** To assess the role of spectral decomposition in weights, activations, and gradients, we replace it with direct FP4 quantization for each case. As shown in Table 2, removing spectral decomposition from gradients yields the largest degradation relative to Metis, with training loss increasing by 2.4% and downstream performance dropping by an average of 1.0%. In comparison, removing it from activations or weights results in smaller and similar degradations: training loss rises by about 0.5% and downstream performance decreases by an average of 0.3%.

**Sparse Random Sampling.** We further evaluate Sparse Random Sampling by replacing it with full-batch spectral decomposition. As shown in Table 2, sparse random sampling reduces the decomposition cost by orders of magnitude while introducing negligible performance impact.

Table 2: Performance of LLaMA-3 8B under different ablation settings of Metis. "w/o" denotes removal of the corresponding component. For FP4 training, spectral decomposition is most critical for gradients (removal yields the largest performance loss), followed by activations and weights. Sparse random sampling performs on par with full-batch spectral decomposition.

| Method | Loss | ARC-C | ARC-E | BoolQ | LAMBADA | PIQA | RACE | HellaSwag | Avg |
|---|---|---|---|---|---|---|---|---|---|
| Metis | 2.45 | 32.4 | **60.4** | **59.2** | 37.6 | 70.5 | **47.0** | 50.9 | **51.1** |
| w/o Weight Decomposition | 2.47 | 32.7 | 58.6 | 58.5 | 36.9 | 71.0 | 45.8 | **51.4** | 50.7 |
| w/o Activation Decomposition | 2.46 | **33.9** | 59.2 | 58.7 | 35.8 | **71.6** | 46.2 | 51.2 | 50.9 |
| w/o Gradient Decomposition | 2.51 | 30.6 | 60.3 | 59.1 | 36.2 | 68.5 | 46.8 | 49.7 | 50.1 |
| w/o Sparse Random Sampling | **2.44** | 32.9 | 60.0 | 59.1 | **37.8** | 70.7 | 46.7 | 50.5 | **51.1** |

## 4.3 EXTENDED EVALUATION

Metis is designed as a training-time method to address the compute and memory bottlenecks of high-precision LLM training such as BF16. We therefore focus primarily on FP4 pre-training results, and additionally evaluate a lightweight inference-time variant under post-training quantization (PTQ) for completeness.

**Pre-training Comparison with Nvidia's Recipe.** We compare Metis with Nvidia's FP4 recipe (Devleker & Ghodsian), which incorporates a random Hadamard transform alongside SR to mitigate the impact of outliers. Our implementation of this recipe is described in Appendix C.1. As shown in Table 3 for GPT-2 models, Nvidia's recipe results in 1-2% higher test loss and an average 0.5% drop in downstream performance relative to the BF16 baselines, whereas Metis surpasses BF16 on both test loss and average downstream performance.

Table 3: Nvidia's recipe yields 1-2% higher test loss and a 0.5% drop in downstream performance relative to BF16, whereas Metis surpasses BF16 on both metrics.

| Model | Method | Loss | ARC-C | ARC-E | BoolQ | LAMBADA | PIQA | RACE | HellaSwag | Avg |
|-------|--------|------|-------|-------|-------|---------|------|------|-----------|-----|
| GPT-2 | BF16 | 3.23 | **24.3** | **32.1** | **60.4** | 31.3 | **62.1** | 47.3 | **32.5** | 41.4 |
| (130M) | FP4-Metis | **3.20** | 24.1 | 31.9 | 60.2 | **31.8** | 61.8 | **48.4** | 32.2 | **41.5** |
| | FP4-Nvidia's Recipe | 3.27 | 23.7 | 31.2 | 59.4 | 30.5 | 62.0 | 47.6 | 31.9 | 40.9 |
| GPT-2 | BF16 | 3.09 | **30.7** | 59.5 | **60.0** | 35.4 | **64.9** | 47.3 | 41.1 | 48.4 |
| (1.1B) | FP4-Metis | **3.01** | 30.1 | **60.0** | 59.5 | 36.2 | 64.1 | **48.5** | **41.9** | **48.6** |
| | FP4-Nvidia's Recipe | 3.15 | 28.6 | 59.8 | 59.3 | **36.7** | 63.2 | 48.0 | 40.2 | 47.9 |

The advantage of Metis lies in two aspects: (i) as shown in A.3, spectral decomposition yields much narrower residual distributions, whereas random Hadamard transform only smooths a few outliers without reducing overall spread; and (ii) as shown in Fig. 6, Metis attains better alignment of singular directions and lower singular-value error than both direct NVFP4 and NVFP4 with Nvidia's recipe, thereby preserving the spectral structure critical for model performance.

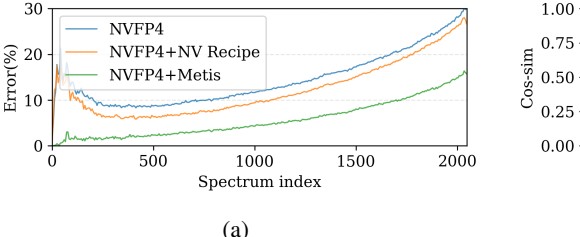
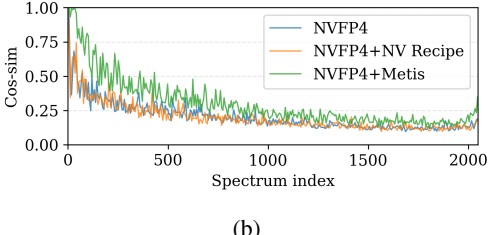

(a)  (b)

Figure 6: Spectral preservation under different quantization strategies. (a) Alignment of left singular vectors measured by cosine similarity. (b) Relative error in singular values. Metis achieves the highest vector alignment and the lowest singular-value error relative to the BF16 baseline.

**PTQ Comparison.** We evaluate a lightweight inference variant of Metis on the LLaMA-3 8B model and compare it with strong PTQ baselines including LQ-LoRA (Guo et al., 2024) and OmniQuant (Shao et al., 2023). At inference time, Metis applies spectral decomposition only to model parameters, eliminating activation-side spectral processing and adding only 3% FLOPs when using a 1.5% rank. As shown in Table 4, Metis achieves higher downstream performance than both baselines, with an average improvement of 0.6% over LQ-LoRA and 0.7% over OmniQuant, while remaining close to the BF16 baseline.

Table 4: Metis improves average downstream performance by 0.6% over LQ-LoRA and 0.7% over OmniQuant, while requiring only 3% additional FLOPs and remaining close to the BF16 baseline.

| Model | Method | Loss | ARC-C | ARC-E | BoolQ | LAMBADA | PIQA | RACE | HellaSwag | MMLU | Avg |
|-------|--------|------|-------|-------|-------|---------|------|------|-----------|------|-----|
| LLaMA-3 | BF16 | **2.44** | 32.4 | **60.4** | **59.2** | **37.6** | 70.5 | **47.0** | **50.9** | 30.7 | **48.6** |
| (8B) | LQ-LoRA | 2.51 | 32.6 | 60.1 | 58.4 | 34.9 | **71.3** | 45.5 | 48.6 | 29.4 | 47.7 |
| | OmniQuant | 2.47 | 30.9 | 59.4 | 58.1 | 36.8 | 69.3 | 46.3 | 48.7 | 31.0 | 47.6 |
| | Metis | 2.45 | **33.2** | 59.2 | 59.0 | 37.3 | 69.9 | 46.9 | 49.5 | **31.3** | 48.3 |

## 5 RELATED WORKS

Block-wise microscaling alleviates FP4's range mismatch but still compresses small values into few bins, causing information loss. Existing methods fall into three categories:

**Channel-wise Re-parameterization.** Methods like SmoothQuant (Xiao et al., 2023) use calibrated per-channel scaling to balance activation and weight magnitudes, folding scales offline. Outlier Suppression+ (Wei et al., 2023) adds per-channel shifts, while OmniQuant (Shao et al., 2023) introduces learnable transforms and weight clipping. These diagonal methods struggle with residual extremes in low-bit settings.

**Hadamard transformations.** Orthogonal Hadamard rotations (Suresh et al., 2017) redistribute outliers across channels to relax per-block ranges. QuaRot (Ashkboos et al., 2024) applies them to hidden states and weights for end-to-end 4-bit inference, while QuIP (Chee et al., 2023) uses randomized preprocessing for weight-only 4-bit PTQ. HALO (Ashkboos et al., 2025) inserts rotations in both forward and backward passes to stabilize low-precision training. NVIDIA's FP4 recipe (De-vleker & Ghodsian) similarly combines random Hadamard transforms with stochastic rounding for W4A4G4 training. Despite these advances, all Hadamard-based methods incur notable overhead from repeated rotations and mainly smooth outliers without narrowing overall distributions, leaving quantized tensors vulnerable to precision loss.

**Outlier Separation.** These methods divert extreme values into a small high-precision branch. Outlier Clamping and Compensation (Wang et al., 2025) clamps top-quantile activations and recovers clipped residuals via sparse GEMM, while SVDQuant (Li et al., 2024) shifts activation mass into weights and absorbs outlier energy in a high-precision low-rank branch. Both reduce error but rely on auxiliary precision paths, deviating from full FP4.

These approaches fall short for end-to-end FP4 training: channel-wise methods are calibration-dependent, Hadamard rotations add complexity without ensuring uniform variance, and outlier separation requires high-precision branches. Instead, we use randomized SVD to disentangle dominant and long-tail singular vectors, transforming broad distributions into quantization-friendly, narrow-range forms with minimal overhead, enabling fully FP4 training.

## 6 CONCLUSIONS

We identified anisotropy in the singular value spectra of parameters, activations, and gradients as a fundamental obstacle to low-bit LLM training and introduced *Metis*, a spectral-domain quantization framework that mitigates this challenge by partitioning spectra into narrower sub-distributions and preserving structural fidelity with negligible overhead. On LLaMA-3 8B, Metis enables robust W4A4G4 training with less than 0.4% loss gap and under 0.1% downstream degradation relative to BF16, while also surpassing Nvidia's FP4 recipe. These results establish Metis as a practical, high-fidelity approach for efficient FP4 training of large language models. Ongoing work explores scaling to larger models and hardware-native deployment, with the goal of broadening access to high-performance, energy-efficient LLMs.

ACKNOWLEDGMENTS

This work was supported by Fudan University and the Shanghai Innovation Institute. Partial support for this work was also provided to Yujiang Wang through the Basic Research Program of Jiangsu (BK20240414) and the Leadership Talent Program (Science and Education) of Suzhou Industrial Park (KJQ2024204).

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

APPENDIX

This section is organized to present empirical results on additional model structures and modules, which complement the *Analysis* and *Methods* sections, as well as experimental details and supplementary results that support the *Experiments* section.

# A ANALYSIS

## A.1 ANISOTROPY: A UNIVERSAL PROPERTY OF MODERN LLMS

The universality of anisotropy is further supported by evidence from open-sourced LLM weight matrices.

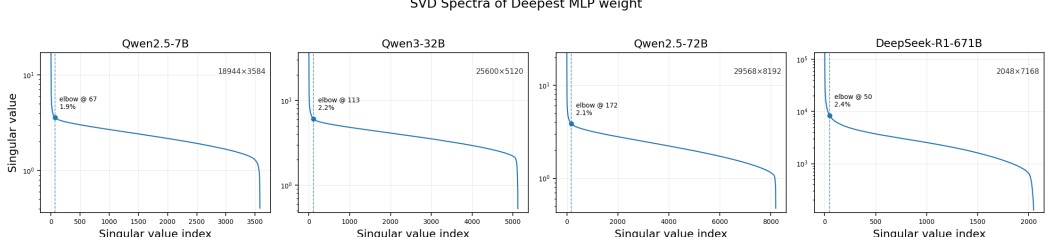

Figure 7: Singular value spectra of the final FeedForward module in Qwen2.5-7B, Qwen3-32B, Qwen2.5-72B, and DeepSeek-R1-671B. The elbow fraction $f = k^\star/r$, where $k^\star$ is the index of maximum curvature, indicates that only a small fraction of singular values dominates the spectrum (1.9%, 2.2%, 2.1%, 2.4%), demonstrating the universality of anisotropy in modern LLMs.

### A.1.1 LLAMA-3 8B

Fig. 2 demonstrated that weight, activation, and gradient matrices exhibit highly anisotropic spectra, with only a few singular values carrying most of the energy. To further substantiate this observation, we provide additional results for LLaMA-3 8B in Figures below. Specifically, we show the singular value spectra and matrix distributions of $W_k$ and $W_{ffn1}$ from the first layer, as well as from the last layer. Consistent with the main analysis, these spectra are strongly anisotropic: the leading singular values dominate, while the majority of smaller components contribute values concentrated near zero. These results confirm that spectral anisotropy is a persistent phenomenon across different layers and parameter types.

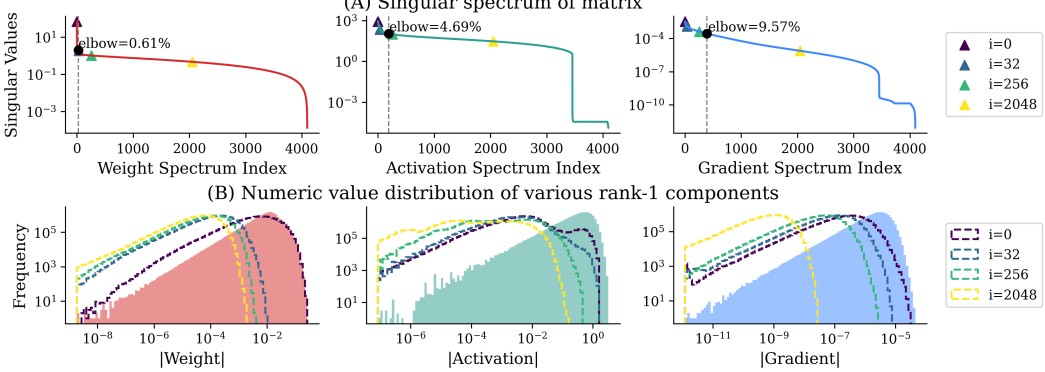

Figure 8: Analysis of weight, activation, and gradient matrices (layer 1, Attention Key). Singular value spectra show strong anisotropy, with a few values carrying most of the energy. Dominant components drive the high-value region, while smaller ones contribute near zero.

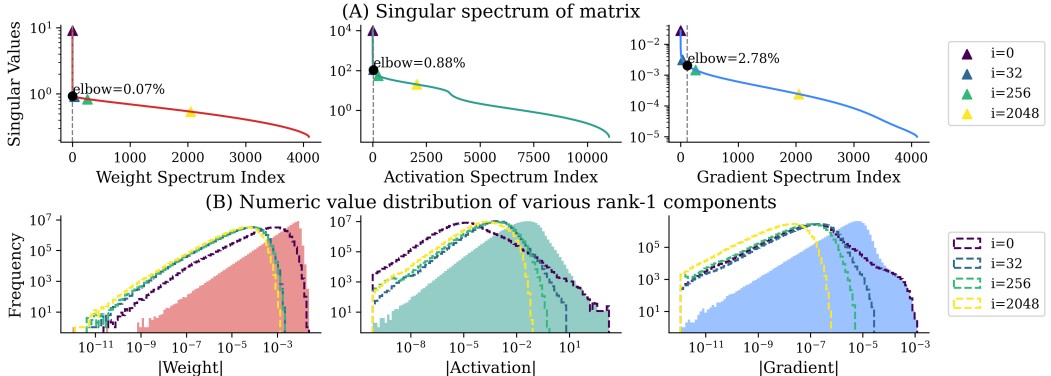

Figure 9: Analysis of weight, activation, and gradient matrices (layer 1, FFN dense2). Singular value spectra show strong anisotropy, with a few values carrying most of the energy. Dominant components drive the high-value region, while smaller ones contribute near zero.

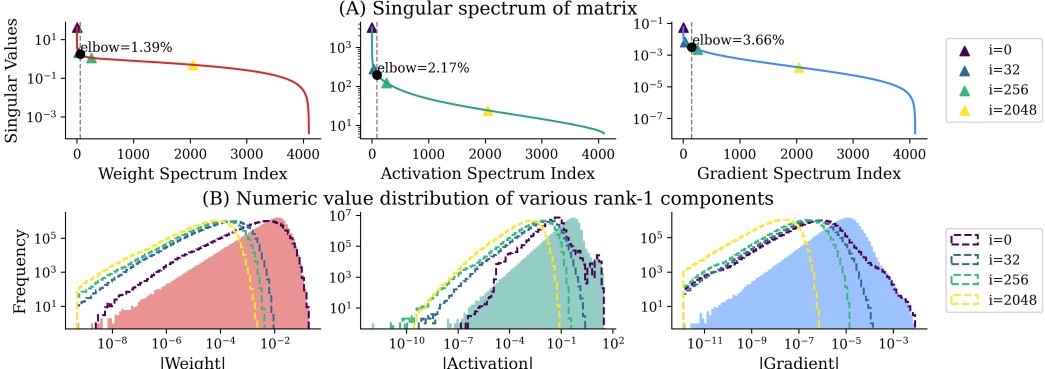

Figure 10: Analysis of weight, activation, and gradient matrices (layer 32, Attention Key). Singular value spectra show strong anisotropy, with a few values carrying most of the energy. Dominant components drive the high-value region, while smaller ones contribute near zero.

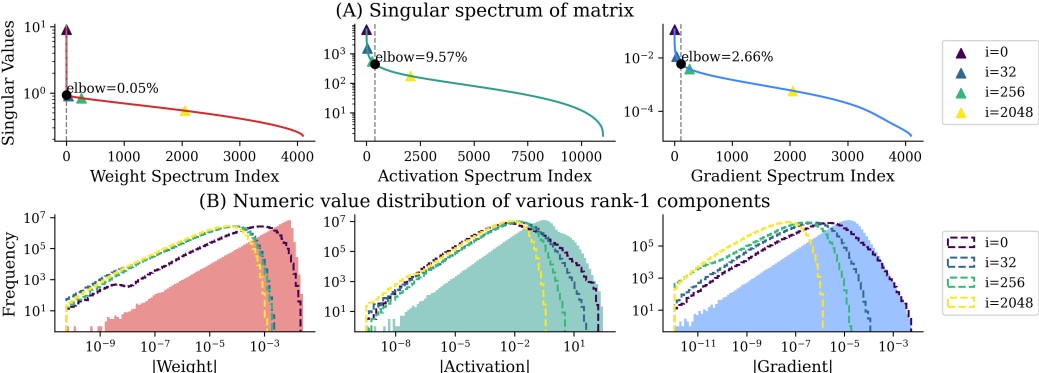

Figure 11: Analysis of weight, activation, and gradient matrices (layer 1, FFN dense2). Singular value spectra show strong anisotropy, with a few values carrying most of the energy. Dominant components drive the high-value region, while smaller ones contribute near zero.

### A.1.2   GPT-2 1.1B

Fig. 2 demonstrated that weight, activation, and gradient matrices exhibit highly anisotropic spectra, with only a few singular values carrying most of the energy. To further substantiate this observation, we provide additional results for GPT-2 8B in Figures below.

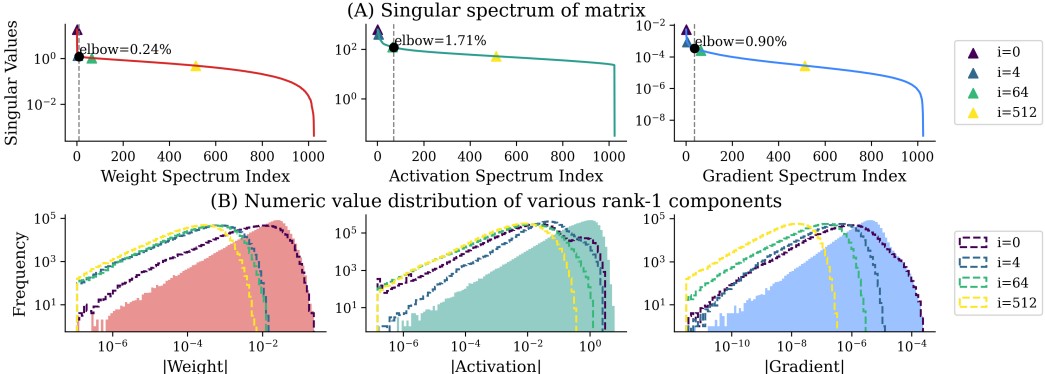

Figure 12: Analysis of weight, activation, and gradient matrices (layer 1, Attention Key). Singular value spectra show strong anisotropy, with a few values carrying most of the energy. Dominant components drive the high-value region, while smaller ones contribute near zero.

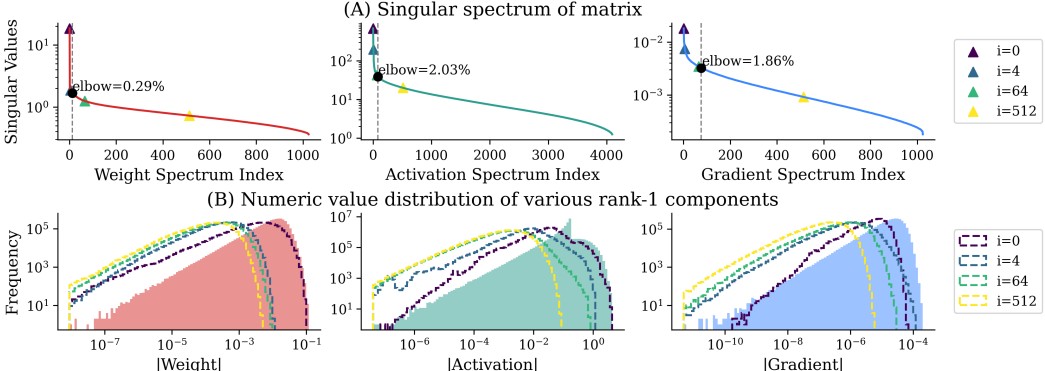

Figure 13: Analysis of weight, activation, and gradient matrices (layer 1, FFN dense2). Singular value spectra show strong anisotropy, with a few values carrying most of the energy. Dominant components drive the high-value region, while smaller ones contribute near zero.

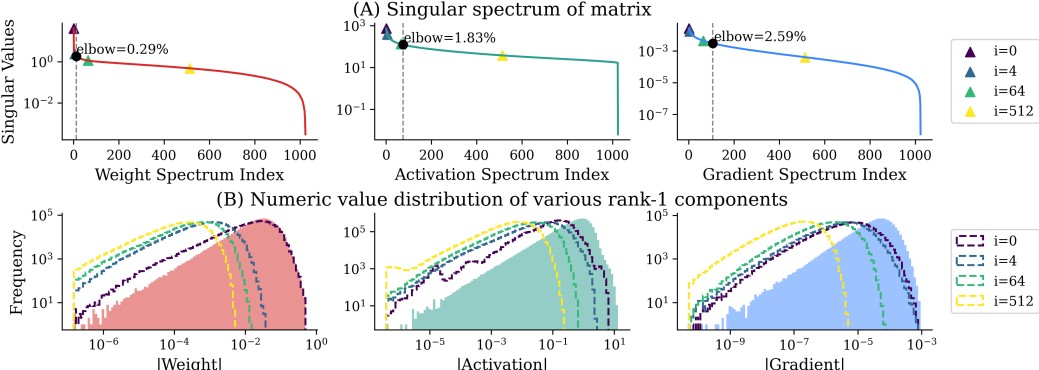

Figure 14: Analysis of weight, activation, and gradient matrices (layer 32, Attention Key). Singular value spectra show strong anisotropy, with a few values carrying most of the energy. Dominant components drive the high-value region, while smaller ones contribute near zero.

## A.2 ANISOTROPY INDUCES WIDE NUMERICAL DISTRIBUTIONS

This section provides additional empirical evidence from a broader range of models and modules, further supporting the analysis in the main text. The results demonstrate that the distributions of singular vectors remain narrow and largely scale-invariant, while the residual components are substantially compressed after removing the dominant singular values.

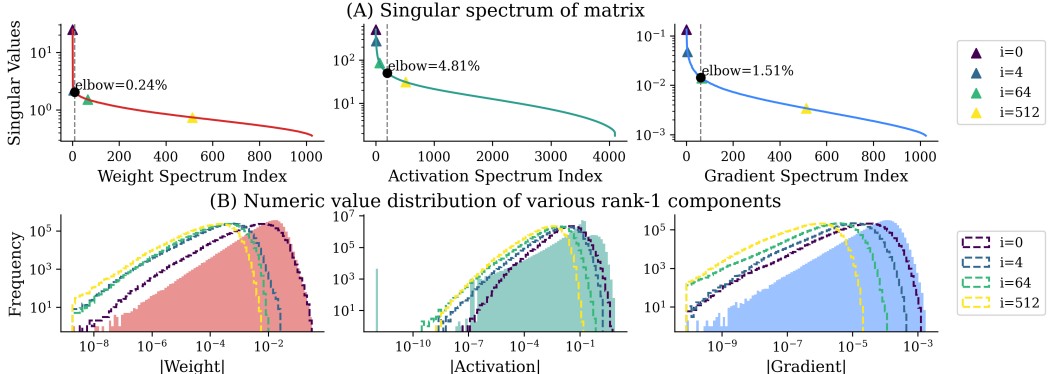

Figure 15: Analysis of weight, activation, and gradient matrices (layer 1, FFN dense2). Singular value spectra show strong anisotropy, with a few values carrying most of the energy. Dominant components drive the high-value region, while smaller ones contribute near zero.

### A.2.1  LLAMA-3 8B

We include further results for LLaMA-3 8B, specifically from layer 1 and layer 32, as well as from the Key projection in the Attention module and the second dense layer in the Feed-Forward Network (FFN).

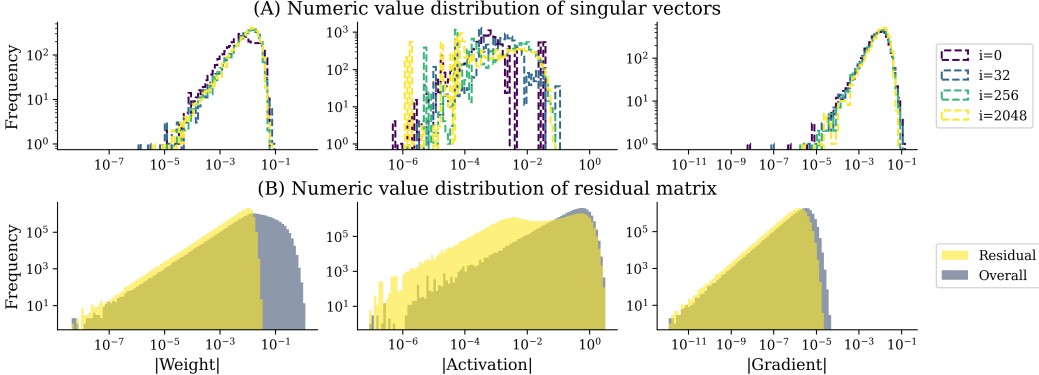

Figure 16: Analysis of weight, activation, and gradient matrices (layer 1, Attention Key), showing that singular vectors are narrow and residuals 1–2 orders of magnitude smaller, confirming wide ranges arise from dominant components amplified by large singular values.

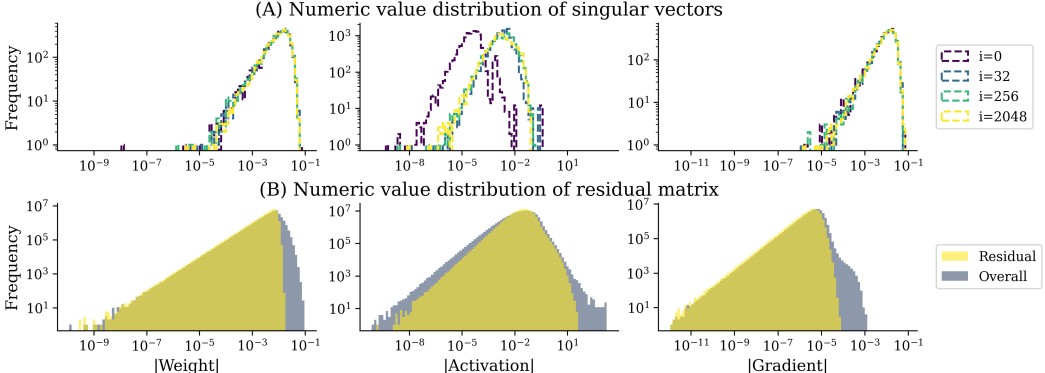

Figure 17: Analysis of weight, activation, and gradient matrices (layer 1, FFN dense2), showing that singular vectors are narrow and residuals 1–2 orders of magnitude smaller, confirming wide ranges arise from dominant components amplified by large singular values.

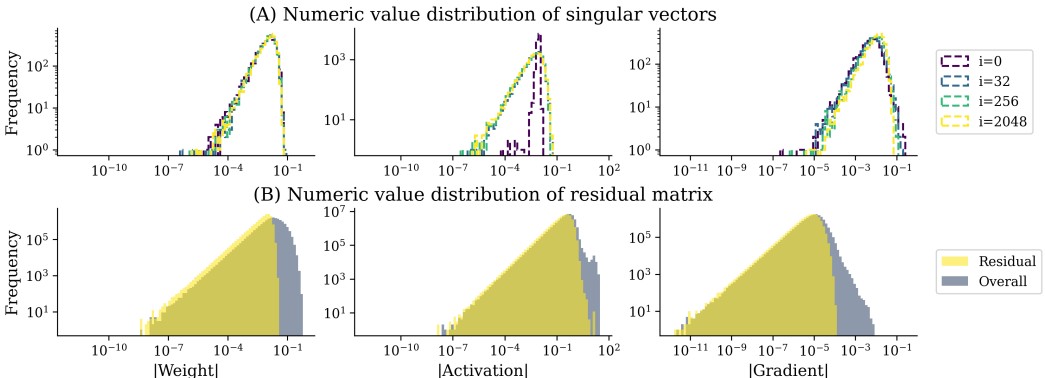

Figure 18: Analysis of weight, activation, and gradient matrices (layer 32, Attention Key), showing that singular vectors are narrow and residuals 1–2 orders of magnitude smaller, confirming wide ranges arise from dominant components amplified by large singular values.

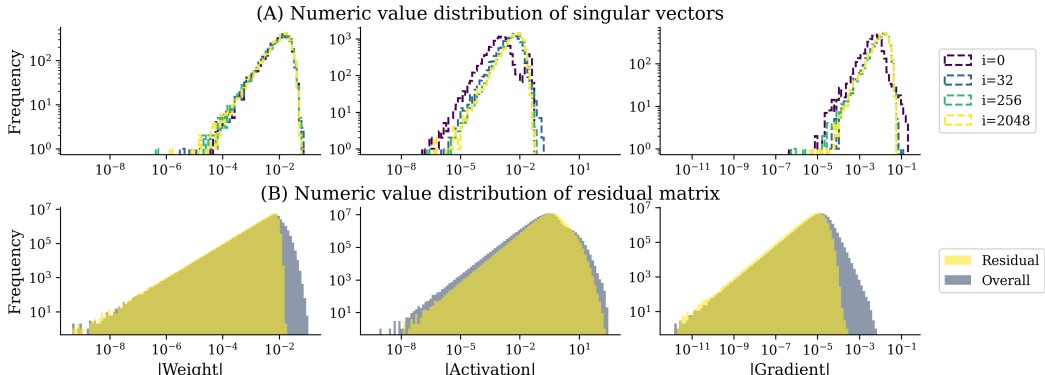

Figure 19: Analysis of weight, activation, and gradient matrices (layer 32, FFN dense2), showing that singular vectors are narrow and residuals 1–2 orders of magnitude smaller, confirming wide ranges arise from dominant components amplified by large singular values.

### A.2.2 GPT-2 1.1B

We include further results for GPT-2 1.1B, specifically from layer 1 and layer 32, as well as from the Key projection in the Attention module and the second dense layer in the Feed-Forward Network (FFN).

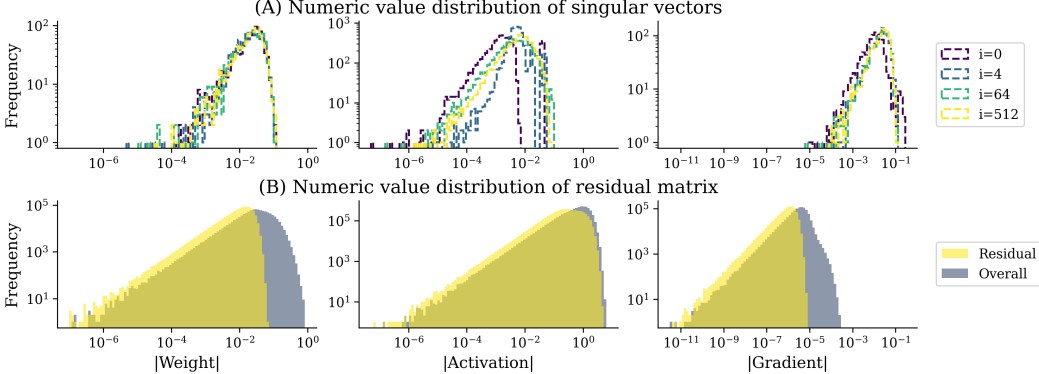

Figure 20: Analysis of weight, activation, and gradient matrices (layer 1, Attention Key), showing that singular vectors are narrow and residuals 1–2 orders of magnitude smaller, confirming wide ranges arise from dominant components amplified by large singular values.

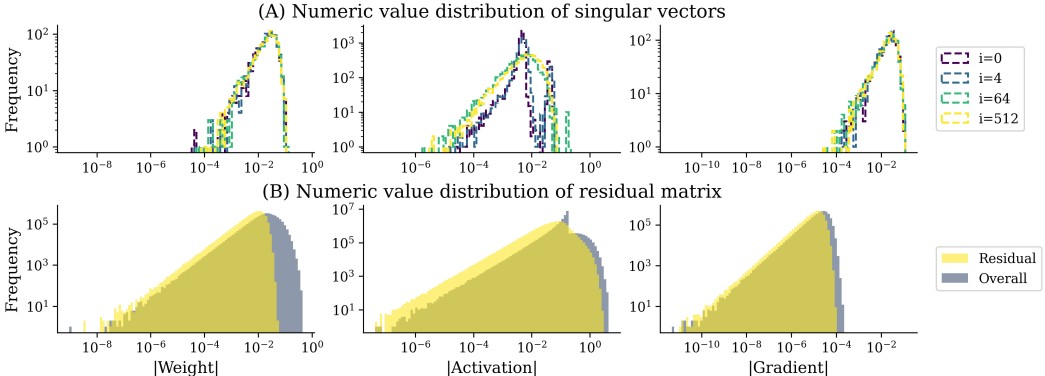

Figure 21: Analysis of weight, activation, and gradient matrices (layer 1, FFN dense2), showing that singular vectors are narrow and residuals 1–2 orders of magnitude smaller, confirming wide ranges arise from dominant components amplified by large singular values.

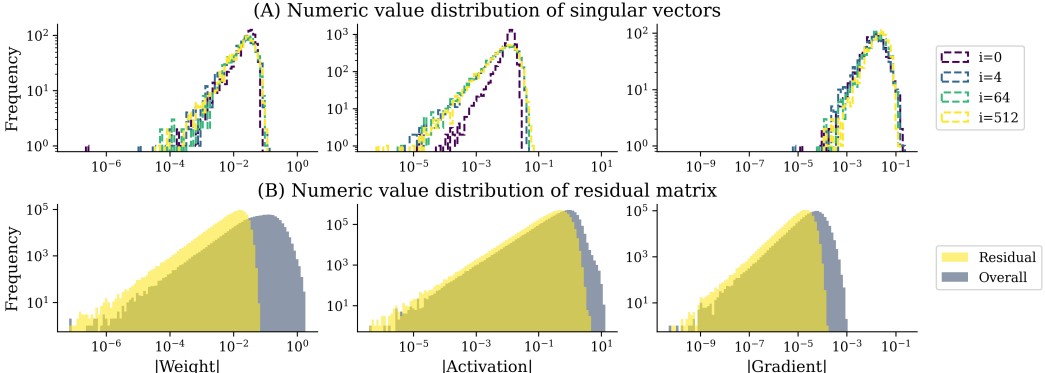

Figure 22: Analysis of weight, activation, and gradient matrices (layer 32, Attention Key), showing that singular vectors are narrow and residuals 1–2 orders of magnitude smaller, confirming wide ranges arise from dominant components amplified by large singular values.

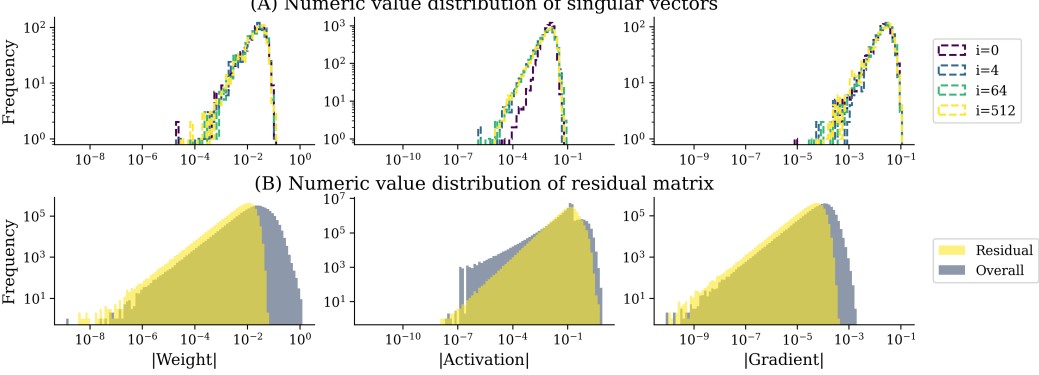

Figure 23: Analysis of weight, activation, and gradient matrices (layer 32, FFN dense2), showing that singular vectors are narrow and residuals 1–2 orders of magnitude smaller, confirming wide ranges arise from dominant components amplified by large singular values.

## A.3 NUMERICAL DISTRIBUTION COMPARISON OF HADAMARD AND METIS

We compare the effects of Hadamard transforms and Metis in the element space. As shown in Fig. 24, the Hadamard transform redistributes only a small fraction of outliers, modestly smoothing the tails but leaving the overall distribution wide and still misaligned with FP4's narrow representable range. In contrast, Metis applies spectral decomposition to separate dominant singular directions and values;

the residual matrix after removing these components exhibits a substantially narrower distribution, inherently more compatible with low-bit quantization.

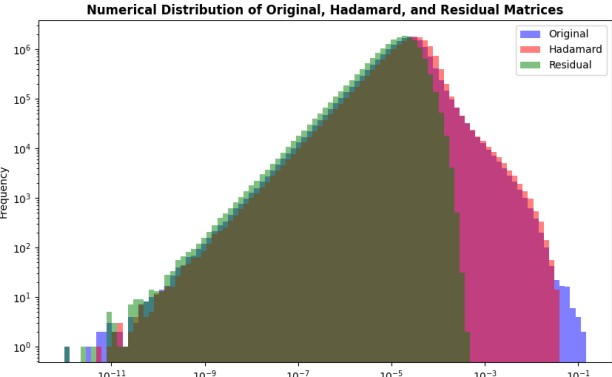

Figure 24: Element-wise distributions under different preprocessing strategies: (i) original tensor, (ii) Hadamard transform, and (iii) Metis spectral decomposition. Hadamard smooths a few outliers but leaves a wide spread, while Metis isolates dominant components and produces a narrower residual distribution well suited for FP4 quantization.

# B    METHOD

## B.1    SPARSE RANDOM SAMPLING SUBSPACE APPROXIMATION

This section provides results on additional modules, further supporting the findings in the *Methods* section and showing that the dominant subspace of a large batch can be efficiently approximated using only a subset of samples.

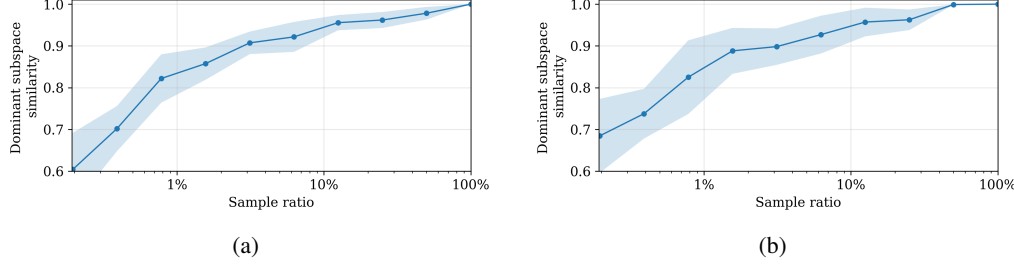

Figure 25: Subspace alignment between the dominant subspace of the full batch and that of randomly sampled subsets of sequences. (a) The input activation of $W_k$ module of layer 32. (b) The input activation of FFN module of layer 32. Alignment quickly saturates as the sample ratio increases, with just 1% of sequences achieving nearly 0.9 alignment with the full-batch subspace.

## B.2    OPERATOR-LEVEL OPTIMIZATION OF SPECTRAL DECOMPOSITION.

To further reduce the practical overhead of repeated randomized SVD, we replace the generic `torch.svd_lowrank` path with a specialized *two-stage pipelined eig-SVD* implementation tailored to our training setting (CUDA, small target rank, and $n_{\text{iter}} = 1$). Instead of performing a full SVD on the compressed matrix $B = Q^\top A$, we form the small Gram matrix $G = BB^\top \in \mathbb{R}^{q \times q}$, compute its eigendecomposition $G = U_e \operatorname{diag}(S^2) U_e^\top$, and then recover the singular vectors as

$$U = QU_e, \qquad V = B^\top U_e \operatorname{diag}(S^{-1}).$$

This converts the final factorization into a $q \times q$ eigensolve, significantly reducing the cost of the decomposition tail.

Table 5: Runtime breakdown of Metis quantization. Overhead is measured relative to the NVFP4 backward baseline.

| Operation | Time (ms) | Overhead vs. NVFP4 |
|---|---|---|
| NVFP4 backward (baseline) | 5.43 | 0% |
| BF16 backward | 9.35 | 72.19% |
| *Spectral step components* | | |
| Spectral decomposition (random sampling) | 0.732 | 13.48% |
| GeMM of $US$ | 0.587 | 10.81% |
| GeMM of $USV$ | 0.253 | 4.66% |
| Quantization of low-rank branch | 0.359 | 6.61% |
| Metis quantization | 1.931 | 35.57% |
| *Amortized cost (1/8 spectral frequency)* | | |
| Metis quantization (amortized) | 0.241 | 4.45% |

The implementation is organized into a front graph and a back graph. The front graph performs randomized range finding and basis construction, including $Y = AR$, one power iteration, and a Cholesky-QR orthogonalization, and then builds $B$ and $G$. The back graph reconstructs $U$ and $V$ from the eigensystem. To reduce launch and synchronization overhead, both stages are captured with CUDA Graphs and reuse persistent device buffers. In addition, the small matrix $G$ is copied to pinned host memory, and its eigendecomposition is executed through a dedicated CPU worker thread. This design moves the small eigensolve off the main GPU factorization path and exposes a pipelined interface for overlapping the CPU-side eigensolve with surrounding work when needed. We further cache the graph executor by shape, device, and dtype, avoiding repeated memory allocation and graph recapture across iterations. Together, these optimizations substantially reduce the wall-clock overhead of the spectral decomposition operator in Metis.

### B.3 RUNTIME EVALUATION

We evaluate the kernel-level wall-clock overhead of Metis on an NVIDIA B300 GPU. Compared to the NVFP4 baseline, Metis introduces an overall runtime overhead of approximately 10.12%. This additional cost comes from two sources: (1) spectral decomposition-based quantization, which contributes 4.45% amortized overhead, and (2) the extra low-rank matrix multiplications introduced by the spectral branch, which contribute 5.67% overhead.

**Setup and Overall Breakdown.** To analyze the runtime overhead in a representative setting, we benchmark the backward pass of a linear module with weight shape $[2048, 2048]$ and input shape $[256, 2048, 2048]$, which matches typical matrix sizes in practical LLM training. A single Metis quantization step incurs an additional 1.93 ms over the NVFP4 backward baseline due to spectral decomposition, low-rank quantization, and the associated low-rank GeMMs.

**Amortized Spectral Decomposition Cost.** The decomposition overhead can be substantially reduced through temporal reuse. With the Sparsely Random Sampling strategy, the dominant low-rank spectral components can be reused across gradient-accumulation steps. Since the dominant subspace estimated from only 1% of the data already achieves approximately 0.9 similarity to the full-data subspace (Fig. 4), we use a reuse interval of 8 steps without observable accuracy degradation. As a result, the decomposition-related overhead is amortized from 1.93 ms per spectral update to 0.24 ms per training step, corresponding to only 4.45% overhead relative to NVFP4.

**Low-Rank Branch Overhead and Kernel Fusion.** Beyond the decomposition itself, Metis also introduces extra low-rank matrix multiplications. To reduce their runtime impact, we design a kernel-fusion strategy that minimizes additional GMEM traffic from the low-rank branch. Specifically, we: (1) load the input tile $X$ from GMEM into SMEM only once; (2) load the corresponding low-rank factor tiles together with the residual branch tiles into SMEM in the same stage; and (3) perform the fused computation entirely in SMEM, accumulate partial results locally, and write the final

Table 6: Overhead breakdown of the fused low-rank branch.

| Operation | Overhead |
|---|---|
| Low-rank parameter reads | 0.74% |
| SMEM writes | 0.62% |
| Computation | 4.30% |
| Others | 0.01% |
| Overall overhead | 5.67% |

output back to GMEM only once. This avoids redundant global-memory reads and writes that would otherwise dominate the cost of the small low-rank branch.

After fusion, the extra low-rank matrix multiplications contribute only 5.67% total overhead. As shown in Table 6, most of this cost comes from computation itself, while memory traffic and other auxiliary costs remain minor.

Overall, these results show that the additional cost of Metis remains modest in practice. By combining sparse random sampling, temporal reuse of spectral components, and kernel fusion for the low-rank branch, Metis keeps the total runtime overhead to 10.12% while preserving the training benefits of spectral decomposition-based quantization.

## C  EXPERIMENTS

### C.1  IMPLEMENTATION OF NVFP4 AND NVIDIA'S RECIPE

In our implementation, the NVFP4 format is simulated by explicitly casting tensors to FP4 before invoking low-precision operators and subsequently restoring them to higher precision when passing results to subsequent high-precision operators. Concretely, the quantization function maps the scaled values of each block into the discrete representable set $\pm\{0, 0.5, 1, 1.5, 2, 3, 4, 6\}$. This process ensures that the simulated tensor values adhere to the constraints of the FP4 e2m1 format, thereby capturing the representational limitations of the hardware specification. By alternating between quantized (FP4) and restored (higher-precision) states, the simulation reproduces the effective numerical behavior of NVFP4 operators while remaining compatible with standard high-precision computational routines.

**Rationale**  The soundness of this simulation stems from the design of FP4 hardware multiply units, which typically retain extra exponent headroom when computing intermediate products. This architectural property guarantees that low-precision multiplication does not incur overflow, even though the operands are quantized. Consequently, performing multiplications in higher precision faithfully reflects the outcome of true FP4 multipliers, since no additional rounding error or overflow is introduced in the product stage. Following the multiplication, accumulation is conducted in bfloat16 (bf16) precision, which aligns with hardware practice and preserves numerical consistency with higher-precision accumulation. Taken together, these considerations indicate that the proposed simulation of NVFP4 matrix multiplications is both practical and theoretically well-justified.

### C.2  SENSITIVITY ANALYSIS OF RANK

Sensitivity analysis of spectral decomposition rank ranging from 1.5% to 12.5%. The curves for 1.5% and 12.5% closely match, indicating that a rank of 1.5% is sufficient to maintain performance.

### C.3  ISOTROPY OF THE RESIDUAL BRANCH

We inspect the singular spectrum of a residual matrix from GPT-2 trained with Metis to examine whether anisotropy re-emerges, using a baseline-trained matrix for comparison. As shown below, the baseline matrix exhibits strong anisotropy, with a few singular values dominating the spectrum, whereas the residual shows a much flatter distribution, indicating that anisotropy is effectively addressed by the low-rank branch in Metis.

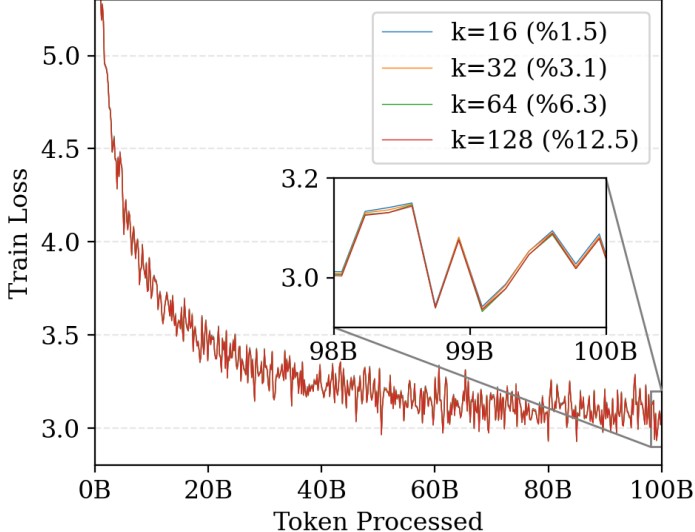

Figure 26: Training loss curves of GPT-2 1.1B using different ranks in spectral decomposition, showing that a rank of 1.5% is sufficient to maintain performance.

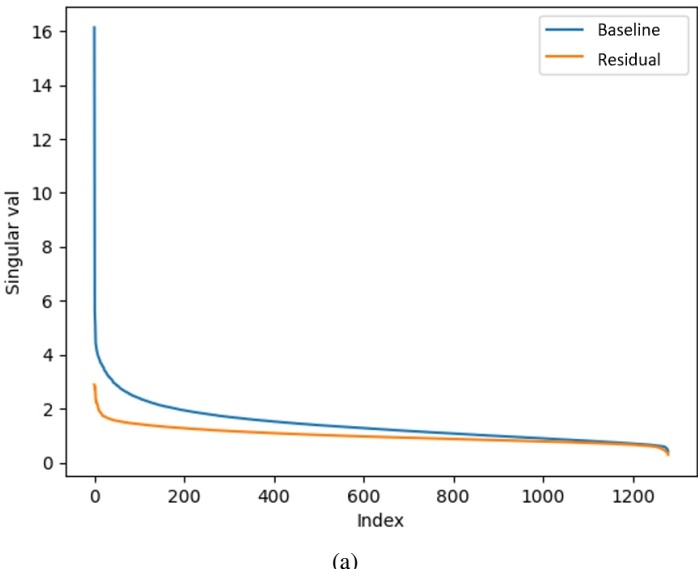

(a)

Figure 27: Inspection of residual anisotropy in Metis compared with a baseline-trained matrix (layer 16, FFN dense1, GPT-2 1.1B). While the baseline matrix exhibits strong anisotropy, the residual spectrum shows a much flatter distribution, indicating that anisotropy is effectively addressed by the low-rank branch in Metis.

