# OpenReview forum: "Metis: Training LLMs with FP4 Quantization"
_ICLR.cc/2026/Conference — ICLR 2026 Poster_

### Official Review · Reviewer_vMTK · 2025-10-27

**Soundness:** 3
**Presentation:** 3
**Contribution:** 2
**Rating:** 6
**Confidence:** 4

**Summary:**

This paper proposes Metis, a spectral-domain quantization framework for FP4 training of large language models. It identifies anisotropy in the singular value spectra of weights, activations, and gradients as a key barrier to low-bit training. Metis partitions these anisotropic spectra into narrower sub-distributions for independent quantization, greatly reducing distortion and preserving spectral structure. It introduces efficient decomposition techniques—sparse random sampling and random projection—to minimize overhead. Experiments on GPT-2 (130M, 1.1B) and LLaMA-3 (8B) show that Metis achieves only 0.4% higher loss and 0.1% accuracy degradation relative to BF16, outperforming NVIDIA’s FP4 recipe.

**Strengths:**

Addresses a fundamental bottleneck (anisotropy) in ultra-low-bit LLM training.

The spectral-domain perspective is novel and well-motivated.

Theoretical analysis and empirical evidence are solid and clearly explained.

The proposed techniques (sparse sampling, random projection) make FP4 training computationally feasible.

Demonstrates strong results on multiple models, surpassing existing FP4 baselines.

**Weaknesses:**

Evaluation is limited to up to 8B models. Since anisotropy worsens with scale, validating on 70B+ models would better demonstrate scalability.

End-to-end efficiency (e.g., wall-clock speedup, memory usage) is not quantitatively reported; the paper mainly provides theoretical overhead analysis.

More comparison with other recent FP4/low-bit training frameworks (e.g., HALO, SVDQuant) would strengthen the positioning.

Experiments rely on FP4 simulation in BF16, not native FP4 hardware, which limits the practical verification of speed and stability claims.

**Questions:**

Questions

How well does Metis scale to larger models (e.g., 70B or 100B)?

What are the measured training throughput and memory savings compared with NVFP4 or FP8 baselines?

What is the difference bitween the training method of Metis and BitNet-family (binary of tenary training)?

---

> ### Author Response · Authors · 2025-11-21
>
> Thank you very much for your time and valuable comments. We provide the relevant figures in the anonymous link: https://anonymous.4open.science/r/ICLR2026_rebuttal-A48C/, referred to as [LINK] below. We summarize our responses as follows:
>
> - **Scalability to 70B+ models:** Due to the excessive monetary cost and limited time, we cannot train 70B+ models directly, but evidence shows that anisotropy scales consistently from 7B to 671B, and Metis integrates naturally with TP/PP/DP training.
> - **Runtime Analysis:** We have provided wall-clock and kernel-level measurements on NVIDIA Blackwell Ultra (GB300). Metis introduces 10.12% additional cost over NVFP4.
> - **Positioning among FP4/low-bit frameworks:** Although Metis was originally developed to improve training-time efficiency with full FP4 quantization, its lightweight inference variant also surpasses other PTQ baselines. We compare Metis in both pre-training and PTQ settings to demonstrate its superiority.
> - **Difference from BitNet-family training:** BitNet approaches use high-precision gradients, whereas Metis enables true full-FP4 training and successfully addresses the challenge of gradient quantization.
>
> Detailed responses follow.
>
> **Q1: How well does Metis scale to larger models?**
>
> A1: Training 70B+ models is beyond our computational budget in academia. However, we believe Metis scales reliably to larger models for two reasons:
>
> a) Anisotropy scales consistently with model size.
>  Appendix A.1 shows that models from 7B to 671B parameters all exhibit strong anisotropy, with fewer than 3% of singular values dominating the spectrum. This indicates that the phenomenon Metis exploits is common across model sizes.
>
> b) Metis integrates naturally with large-scale parallelism.
>  Metis is naturally compatible with PP and DP, where it is applied locally within each pipeline stage or data-parallel replica. In tensor parallelism, Metis leverages randomized dimensionality reduction: the range-finder GEMMs are run in a distributed fashion, communication happens only on the reduced dimensions, and the final small SVD is computed on one GPU and broadcast to the TP group.
>
> **Q2: Runtime evidence on Blackwell GPUs.**
>
> A2: Compared to the NVFP4 baseline, Metis introduces approximately 10.12% computational overhead.
>
> We evaluated the kernel-level, wall-clock computational overheads of Metis on B300 GPU. The extra computational overhead of Metis originates from two operations: (1) spectral decomposition-based quantizations (**4.45%** extra overhead), and (2) the extra small matrix multiplications from spectral decompositions (**5.67%** extra overhead).
>
> **Detailed analysis & overhead breakdown:** We adopted a linear module (shape: [2048, 2048], input shape: [256, 2048, 2048]), which matches the typical shapes in real LLM training, and benchmarked its backward pass time to illustrate the detailed overhead breakdown of Metis. A step of Metis quantization incurs an extra 1.93 ms from the decomposition and its associated quantization and low-rank GEMMs. With the Sparsely Random Sampling strategy, the same low-rank spectral components can be reused across gradient-accumulation steps. Since the dominant subspace estimated from only 1% of the data already matches the full-data subspace with ≈0.9 similarity (Fig.4 in paper), we select a reuse interval of 8 without loss of accuracy. This amortizes the cost to 0.24 ms per step (4.45% overhead vs. NVFP4).
>
> ### **Quantization Strategy Overhead**
> | Operation | Time (ms) | Overhead vs NVFP4 |
> |-|-:|-:|
> |**NVFP4 backward (baseline)** | **5.43** | **0%** |
> | **BF16 backward** | **9.35** | **72.19%** |
> | --- |  |  |
> | **Spectral step components** |  |  |
> | Spectral decomposition (random sampling) | 0.732 | 13.48% |
> | GEMM of U @ S | 0.587 | 10.81% |
> | GEMM of U @ S @ V | 0.253 | 4.66% |
> | Quantization of low-rank branch | 0.359 | 6.61% |
> | Metis quantization | 1.931 | 35.57% |
> | --- |  |  |
> | **Amortized cost (1/8 spectral frequency)** |  |  |
> | **Metis quantization (amortized)** | **0.241** | **4.45%** |
>
> Furthermore, we optimized the small matrix multiplication operations, which led to an overall overhead of 5.67%. We designed a kernel fusion strategy to minimize the extra GMEM traffic introduced by the low-rank branch. This strategy involves: (1) Loading X from GMEM into SMEM only once per tile; (2) Loading corresponding UV and R tiles into SMEM in the same step; (3) Performing computation in SMEM, accumulating results locally, and writing the final output to GMEM once. After fusion, the low-rank branch adds only 5.67% overhead in total. The overhead breakdown is as follows:
> ### **Extra Small Matrix Multiplication Overhead**
> | Operation| Overhead |
> |-|-:|
> | Low-rank parameter reads  | 0.74%|
> | SMEM writes | 0.62% |
> | Computation| 4.30%|
> | Others | 0.01%|
> | **Overall overhead**| **5.67%** |
> The fused kernel code is included in the link https://anonymous.4open.science/r/Metis-quantization-644B.

---

> > ### Author Response · Authors · 2025-11-21
> >
> > **Q3: More comparison with other recent FP4/low-bit training frameworks (e.g., HALO, SVDQuant) would strengthen the positioning.**
> >
> > A3: We have added comparisons to other FP4/low-bit training frameworks in both the pretraining and PTQ settings. Metis is a training-time method, designed to address the compute and memory bottlenecks of high-precision LLM training (e.g., BF16), and is therefore orthogonal to post-training quantization (PTQ) rather than a competitor to it. The fact that its lightweight inference variant, added only for completeness, still outperforms strong PTQ methods is an additional benefit rather than its primary design goal.
> >
> > **Pretraining:** Under full-FP4 training where weights, activations, and gradients are all quantized to FP4, we have added comparisons against both ‘FP4 All the Way’ and HALO on GPT-2 1.1B. Since both methods are trained to 70B tokens, we evaluate Metis at the same 70B-token checkpoint for a fair comparison. In terms of training loss, Metis consistently achieves lower loss than both methods ([LINK] ""Results of other FQT methods). On downstream tasks, Metis outperforms HALO by 0.6% and FP4 All the Way by 0.9%.
> > | Method | Loss | ARC-C | ARC-E | BoolQ | LAMBADA | PIQA | RACE | HellaSwag | MMLU | Avg. |
> > |--------|------:|------:|------:|------:|--------:|-----:|-----:|-----------:|------:|------:|
> > | BF16 | 3.11 | 28.9±0.5 | 51.6±0.3 | 60.1±0.4 | 34.7±0.6 | 62.9±0.3 | 47.6±0.6 | 38.5±0.3 | 24.9±0.7 | 43.7±0.5 |
> > | FP4 all the way | 3.37 | 26.8±0.3 | 52.8±0.6 | 58.4±0.5 | 34.0±0.5 | 59.4±0.9 | 47.3±0.6 | 37.3±0.6 | 24.6±0.5 | 42.5±0.6 |
> > | HALO | 3.20 | 28.7±0.6 | 50.3±0.6 | 59.4±0.3 | 33.9±0.5 | 60.2±0.6 | 46.7±0.8 | 38.1±0.2 | 25.5±1.0 | 42.8±0.6 |
> > | Metis | 3.08 | 29.4±0.5 | 52.1±0.4 | 60.7±0.7 | 34.5±0.5 | 62.3±0.4 | 47.2±0.5 | 38.3±0.7 | 25.2±0.6 | 43.4±0.5 |
> >
> > **Inference / PTQ setting:** At inference time, Metis applies spectral decomposition only to the parameters and eliminates the need for activation-side spectral processing, adding only 3% FLOPs when using a 1.5% rank. Metis achieves higher inference performance over strong PTQ baselines such as LQ-LoRA and OmniQuant on LLaMA-3(8B) model.
> > | Method | Loss | ARC-C | ARC-E | BoolQ | LAMBADA | PIQA | RACE | HellaSwag | MMLU | Avg. |
> > |--------|------:|------:|------:|------:|--------:|-----:|-----:|-----------:|------:|------:|
> > | BF16 | 2.44 | 32.4 | 60.4 | 59.2 | 37.6 | 70.5 | 47.0 | 50.9 | 30.7 | 48.6 |
> > | LQ-Lora | 2.51 | 32.6 | 60.1 | 58.4 | 34.9 | 71.3 | 45.5 | 48.6 | 29.4 | 47.7 |
> > | OmniQuant | 2.47 | 30.9 | 59.4 | 58.1 | 36.8 | 69.3 | 46.3 | 48.7 | 31.0 | 47.6 |
> > | Metis | 2.45 | 33.2 | 59.2 | 59.0 | 37.3 | 69.9 | 46.9 | 49.5 | 31.3 | 48.3 |
> >
> > **Q4: What is the difference between the training method of Metis and BitNet-family (binary of tenary training)?**
> >
> > A4: The BitNet family represents a class of QAT methods whose primary goal is to train a model whose weights can be quantized to 4-bit or even lower at inference time, thereby reducing inference cost. In contrast, Metis is a solution for full FP4 quantization, aiming not only to produce a 4-bit model for inference but also to accelerate training itself with FP4 arithmetic.
> >
> > In BitNet-style methods, the Straight-Through Estimator (STE) is used for weight quantization during training, but the gradients are still computed in high precision. Using low-precision gradients directly with STE would introduce large quantization errors and destabilize training.
> >
> > Metis addresses this challenge by leveraging the observed anisotropy in weights, activations, and gradients. In particular, Metis performs a low-rank decomposition of the gradient matrices to enable stable quantization of the small-gradient components, making true full-FP4 training feasible.

---

### Official Review · Reviewer_kaek · 2025-10-27

**Soundness:** 3
**Presentation:** 2
**Contribution:** 3
**Rating:** 6
**Confidence:** 4

**Summary:**

This paper introduces Metis, a spectral-domain quantization framework designed to enable robust 4-bit (FP4) training for Large Language Models (LLMs). The authors identify anisotropy—the domination of parameter, activation, and gradient spectra by a few large singular values—as the fundamental barrier to low-bit training. Metis solves this by partitioning the anisotropic spectra into narrower sub-distributions, which are then quantized independently to reduce errors and preserve spectral structure. To make this computationally feasible, Metis uses sparse random sampling and random projection to reduce the cost of spectral decomposition.

**Strengths:**

1. This paper provide comprehensive analysis to gradient, weight, activation based on SVD, reveaing that a few singular values dominating the spectrum.
2. This paper demonstrate that absmax quantization favors larger values, and bring significantly bias to small values.
3. The performance is impressive, achieving nearly lossless performance in NVFP4 training.

**Weaknesses:**

1. This paper focus on FP quantization, but the quantization illustrated in Lines 181 are INT quantization formulation.
2. Two different "X" symbol in line 269 and line 270.
3. The writing in section 3.2 is unclear, it should be emphasize which GeMM are execute in 4-bit and which GEMM are execute in BF16.
4. Metis introduces too much additional operation, though the computational FLOPs is relatively small, the additional memory overhead or IO overhead are not negligible. This should be discussed.

**Questions:**

1. what is the meaning of   $\bar{X} in Eq.(2) ,as well as other symbol with bar? This paper do not give definition for these symbols.
2. Why NVFP4 training can achieve lower loss compared to BF16 training in GPT models?

---

> ### Author Response · Authors · 2025-11-21
>
> Thank you very much for your time and valuable comments. We provide the relevant figures in the anonymous link: https://anonymous.4open.science/r/ICLR2026_rebuttal-A48C/, referred to as [LINK] below. We summarize our responses as follows:
>
> - **Clarification of Quantization:** The rounding in Line 181 is FP4-E2M1 floating-point quantization, not INT quantization.
> - **Notation Consistency:** The two “X” symbols represent tensors of different shapes: the earlier calligraphic $\mathcal{X}$ is a 3D tensor, while $X$ is its 2D flattened form. Bar symbols consistently denote quantized matrices.
> - **Precision of GEMM Operations:** GEMMs involving singular-value matrices and right-singular-vector matrices run in BF16, while all remaining GEMMs run in FP4. These BF16 GEMMs contribute only about 0.0001% of total FLOPs.
> - **Overhead of Memory I/O:** Although Metis introduces additional steps, kernel fusion removes redundant memory traffic. The total overhead after fusion is only 5.67%, of which only 1.36% comes from memory overhead.
> - **NVFP4 vs. BF16 Loss Behavior:** We believe Metis slightly outperforms BF16 because the low-rank + residual decomposition better preserves and updates tail components that BF16 tends to under-represent.
>
> Detailed responses are provided below.
>
> **Q1: INT quantization vs FP quantization (Line 181).**
>
> A1: The rounding operator in Line 181 refers to rounding each element within a block to the nearest FP4-E2M1 representable value, after scaling the elements within each block by the maximum value of that block.
>
> **Q2: Two different "X" symbols in line 269 and line 270.**
>
> A2: The calligraphic $\mathcal{X}$ denotes a 3D input activation tensor $\mathcal{X} \in \mathbb{R}^{b \times s \times m}$, where $b$ is the batch size, $s$ the sequence length, and $m$ the hidden dimension. For the matrix computation, $\mathcal{X}$ is reshaped into a 2D matrix $X \in \mathbb{R}^{\ell \times m}$ with $\ell = b \cdot s$.
>
> **Q3: Clarifying which GEMMs use FP4 vs BF16.**
>
> A3: All the GeMM involves the singular value matrix ${T_k}, {S_k}$ and the right singular vector matrix $\overline{Q_k}, \overline{U_k}$ are executed in BF16. For example, in eq.3 that is ${T_k}\overline{Q_k}$and ${S_k}\overline{U_k}$are executed in BF16 and the others are executed in FP4.  These BF16 GEMMs contribute only about 0.0001% of the total FLOPs.
>
> **Q4: Memory / I/O overhead introduced by additional operations.**
>
> A4: We show that kernel fusion can substantially reduce the memory and I/O overhead, resulting in only 5.67% total overhead over the baseline, of which only 1.36% comes from memory overhead.
>
> To assess the memory cost of the additional matrix multiplications, we profiled small-matrix GEMMs using Nsight Compute and found that over 90% of warp stalls arise from global-memory (GMEM) reads/writes.
>
> To mitigate this, we designed a kernel-fusion strategy that eliminates most of the extra GMEM traffic introduced by the low-rank branch. Specifically, when the weight matrix is decomposed into a low-rank part (UV) and a residual part (R), both branches share the same activation input X, and their outputs are summed to produce C. Instead of loading X twice and writing partial results separately, our fused kernel:
> 1. Loads one tile of X from GMEM into SMEM only once.
> 2. Loads the corresponding UV and R tiles into SMEM in the same step.
> 3. Computes both branches in SMEM, accumulates the results locally, and
> 4. Writes the final output C to GMEM only once.
>
> This removes redundant GMEM read/write operations from the low-rank branch.
> After fusion, the entire low-rank branch adds only 5.67% overhead. The overhead breakdown is as follows:
> | Source of Overhead | % |
> |--|---:|
> | Low-rank parameter GMEM reads | 0.74% |
> | SMEM writes | 0.62% |
> | Low-rank compute | 4.30% |
> | Other | 0.01% |
> | **Total** | **5.67%** |
> The fused kernel code is included in the link https://anonymous.4open.science/r/Metis-quantization-644B.
>
> **Q5: Meaning of $\bar{X}$ and other bar-notation symbols.**
>
> A5: Symbols with a bar denote the quantized version of the corresponding matrix, as defined in Section 2.3.
>
> **Q6: Why NVFP4 training can achieve lower loss compared to BF16 training in GPT models?**
>
> A6: We re-ran the 1B experiments with two independent seeds each and we observed that NVFP4 + Metis consistently achieves slightly lower loss values than the BF16 baseline ( [LINK] "Results of muti-seeded experiments").
>
> Our hypothesis is as follows. Even under BF16, the weight and activation distributions can remain overly wide, so small-magnitude (tail) components are still under-represented, and large singular values dominate the updates. In contrast, Metis explicitly decomposes the weight into a low-rank branch and a residual branch that are trained separately, which allows the tail components to be learned more effectively instead of being suppressed by a few large modes. We will investigate this interesting phenomenon more systematically in future work.

---

### Official Review · Reviewer_eipG · 2025-10-30

**Soundness:** 2
**Presentation:** 4
**Contribution:** 2
**Rating:** 2
**Confidence:** 4

**Summary:**

The paper begins by studying the spectral space of weights, activations, and gradients. The findings show a clear "anisotropy" in the singular distribution, which causes wide numerical ranges, hence high low-bit quantization errors. METIS is then designed to 1) decompose the matrices into a low-rank component with large singular values, and a residual component with smaller singular values, 2) do so efficiently using sparse random sampling and random projection, and 3) quantize each component separately. The results show that NVFP4 METIS is able to closely recover BF16 baseline across GPT/Llama models, and outperform naive NVFP4 baseline.

**Strengths:**

1. I believe METIS is generally a very strong paper on the analysis side, and some of their findings are novel and interesting.

2. Very clear presentation.

3. Clean and clear ablation studies, including showing that the residual term in the weights does not "develop" large singular values.

**Weaknesses:**

1. The asymptotic runtime analysis, in the era of Blackwell GPUs, is not convincing at all, especially given all the memory-bound operations (e.g., multiple quantization calls, etc). On the other hand, the authors claim that NVIDIA's FP4 stack was not released at the time of submission, but cutlass already supported NVFP4 matrix multiplication in April 2025 [1]. I think speedup numbers, including layer-wise and end-to-end, are crucial for this paper.

2. Some minor concerns exists, see questions.

[1] https://github.com/NVIDIA/cutlass/releases/tag/v3.9.0

**Questions:**

1. Could the authors provide real-world speedup number? I understand the rebuttal timeframe is too short for writing kernels, but from my point of view, this is critical. The authors can maybe take advantage of the kernels in [2].

2. Several times in the paper it was mentioned that gradients are being quantized. I think this is ambiguous, as it could mean both activation gradients and weight gradients (I believe the authors mean the former). Can you please make this clear?

3. The authors mention that l >> k (around line 222). What will happen if that's not the case? Note that this actually happens quite often, especially in research (with gradient accumulation).

[2] https://github.com/IST-DASLab/qutlass

In general, my biggest concern regarding this paper is about the runtime analysis. If my concern is addressed, I will consider increasing my score, possibly significantly. But I don't think the paper is complete without it.

---

> ### Author Response · Authors · 2025-11-21
>
> Thank you very much for your time and valuable comments. We summarize our responses as follows:
> - **Runtime and Speedup Analysis:** We have provided wall-clock and kernel-level measurements on NVIDIA Blackwell Ultra (GB300). Metis introduces only 10.12% additional cost over NVFP4.
> - **Ambiguity on Gradient Quantization:** We clarify that Metis quantizes both activation gradients and weight gradients.
> - **$l≫k$ Assumption:** We show that $l≫k$ consistently holds for real-world large-scale training. When accumulation is used, Metis reuses low-rank components across accumulation steps.
>
> The detailed responses follow below.
>
> **Q1: Runtime evidence on Blackwell GPUs.**
>
> A1: Compared to the NVFP4 baseline, Metis introduces approximately 10.12% computational overhead.
>
> At submission time, CUTLASS only exposed NVFP4 as low-level GEMM kernels, but did not provide an end-to-end NVFP4 training recipe and forward/backward support for LLM training, which made it infeasible for us to build an NVFP4-based Metis training framework on top of Transformer Engine and Megatron.
>
> We evaluated the kernel-level, wall-clock computational overheads of Metis on B300 GPU. The extra computational overhead of Metis originates from two operations: (1) spectral decomposition-based quantizations (**4.45%** extra overhead), and (2) the extra small matrix multiplications from spectral decompositions (**5.67%** extra overhead).
>
> **Detailed analysis & overhead breakdown:** We adopted a linear module (shape: [2048, 2048], input shape: [256, 2048, 2048]), which matches the typical shapes in real LLM training, and benchmarked its backward pass time to illustrate the detailed overhead breakdown of Metis. A step of Metis quantization incurs an extra 1.93 ms from the decomposition and its associated quantization and low-rank GEMMs. With the Sparsely Random Sampling strategy, the same low-rank spectral components can be reused across gradient-accumulation steps. Since the dominant subspace estimated from only 1% of the data already matches the full-data subspace with ≈0.9 similarity (Fig.4 in paper), we select a reuse interval of 8 without loss of accuracy. This amortizes the cost to 0.24 ms per step (4.45% overhead vs. NVFP4).
>
> ### **Quantization Strategy Overhead**
> | Operation | Time (ms) | Overhead vs NVFP4 |
> |-|-:|-:|
> |**NVFP4 backward (baseline)** | **5.43** | **0%** |
> | **BF16 backward** | **9.35** | **72.19%** |
> | --- |  |  |
> | **Spectral step components** |  |  |
> | Spectral decomposition (random sampling) | 0.732 | 13.48% |
> | GEMM of U @ S | 0.587 | 10.81% |
> | GEMM of U @ S @ V | 0.253 | 4.66% |
> | Quantization of low-rank branch | 0.359 | 6.61% |
> | Metis quantization | 1.931 | 35.57% |
> | --- |  |  |
> | **Amortized cost (1/8 spectral frequency)** |  |  |
> | **Metis quantization (amortized)** | **0.241** | **4.45%** |
>
> Furthermore, we optimized the small matrix multiplication operations, which led to an overall overhead of 5.67%. We designed a kernel fusion strategy to minimize the extra GMEM traffic introduced by the low-rank branch. This strategy involves: (1) Loading X from GMEM into SMEM only once per tile; (2) Loading corresponding UV and R tiles into SMEM in the same step; (3) Performing computation in SMEM, accumulating results locally, and writing the final output to GMEM once. After fusion, the low-rank branch adds only 5.67% overhead in total. The overhead breakdown is as follows:
> ### **Extra Small Matrix Multiplication Overhead**
> | Operation| Overhead |
> |-|-:|
> | Low-rank parameter reads  | 0.74%|
> | SMEM writes | 0.62% |
> | Computation| 4.30%|
> | Others | 0.01%|
> | **Overall overhead**| **5.67%** |
> The fused kernel code is included in the link https://anonymous.4open.science/r/Metis-quantization-644B.
>
> **Q2: Clarify whether ‘gradient quantization’ refers to activation gradients, weight gradients, or both.**
>
> A2: To clarify, Metis quantizes both activation gradients and weight gradients. As described in Section 3.2, the GEMMs used for activation-gradient computation $∂L/∂X$ and those for the weight-gradient computation $∂L/∂U_k, ∂L/∂S_k, ∂L/∂V_k, ∂L/∂W_R$ in both the low-rank and residual branches are all quantized in Metis.
>
> **Q3: The authors mention that l >> k (around line 222). What will happen if that's not the case? Note that this actually happens quite often, especially in research (with gradient accumulation).**
>
> A3: In practical large-scale LLM training, the condition $l≫k$ almost always holds. Here, $l$ is the global batch size multiplied by the sequence length, and it is independent of gradient accumulation. For example, in typical settings such as Llama-8B, $l=512×8192=4,194,304 ≫ k=4096.$
>
> When gradient accumulation is used, Metis reuses low-rank components across accumulation steps.
>
> If $l≫k$  were not to hold, a random projection on the hidden dimension is sufficient to reduce the cost of spectral decomposition. In this regime, random sampling is no longer required.

---

> > ### Author Response · Authors · 2025-11-28
> >
> > We sincerely appreciate your constructive feedback. As the discussion period draws to a close, we would like to follow up to confirm that our responses and the newly added experiments have adequately addressed your concerns. We remain readily available to clarify any remaining questions.
> >
> > If you feel our revisions are satisfactory, we would greatly appreciate your consideration in updating your rating. Your evaluation is important to us and has a meaningful impact on the reception of our work. Thank you again for your time and kind suggestions.

---

### Official Review · Reviewer_mj9Y · 2025-11-01

**Soundness:** 4
**Presentation:** 4
**Contribution:** 3
**Rating:** 4
**Confidence:** 4

**Summary:**

The paper tackles the core problem of why end-to-end quantized (specifically FP4) training of LLMs underperforms. Quantization plus strongly anisotropic tensors (weights, activations, and gradients) leads to disproportionate distortion of the small singular directions. They propose Metis, which does a cheap, sampled spectral decomposition per step, splits tensors into a small low-rank branch and a quantization-friendly residual, and quantizes the resulting narrower parts separately. This gives an almost fully quantized setup (W4A4G4) and, on LLaMA-3 8B pretrained for 100B tokens, the method stays close to BF16 with modest benchmark drops. Experiments are quite strong (8B training, meaningful ablations), and the intuition is clear and well-argued.

**Strengths:**

1. The paper correctly identifies spectral anisotropy (few large singular values + many small ones) as the real reason FP4 fails, and the “separate big directions, then quantize” story is both simple and compelling.

2. Using sequence subsampling and randomized projection to estimate the dominant subspace is a sensible way to make SVD affordable; even if the theoretical bounds are loose, the justification is acceptable for practical use.

3. 8B pretraining at 100B tokens, even if not at full budget, is a significant scale for a quantization paper, and much stronger than GPT-2-only demonstrations.

4. Doing weights, activations, and gradients almost FP4 is a decent advance over weight-only or weight+activation.

5. The ablation that removes spectral decomposition from W/A/G and shows that gradients matter the most is especially convincing and supports their main diagnostic.

6. Modest but consistent benchmark results. The reported downstream gaps are small but believable given the setup.

**Weaknesses:**

1. There is no actual wall-clock or kernel-level overhead comparison, despite the fact that FP4 kernels (NVFP4-style) are already available in systems like QuTLASS and vLLM. Even an FP4 (not NVFP4) timing would have been possible, as it has similar FLOPS properties to NVFP4.


2. At inference time, you still need the fancy activation-side spectral quantization. That limits how much speedup and how much deployment simplicity you actually gain, and it’s unclear how this scales.


3. The experiments are single-seeded, and no CI is provided. Especially for smaller models (GPT-2 scale), results should be multi-seeded; the reported downstream gaps are within typical variance.


4. Benchmarks are modest. Stronger discriminative tasks, such as MMLU, GSM8K, or other more challenging evaluations, would better reveal the gains from this method.


5. No comparison to recent strong PTQ like QTIP, Quip#, AQLM, OmniQuant.

6. No comparison to recent FQT methods, such as “FP4 All the Way” or rotation-based FP4 training (like “HALO” in NVFP4). So it’s hard to situate Metis among existing efforts.


7. The paper says Hadamard incurs notable overhead and doesn’t really fix the distribution, but fast Hadamard is $ O(n \log n) $, and HALO/HALO-like methods do notably reduce kurtosis to a Gaussian level (see HALO §A.4).


8. The method is algebraically described but not presented as a concrete algorithm.


9. It would have been more informative to include validation loss curves instead of or along the training loss curve.


10. The decomposition bounds presented are quite loose; that’s acceptable as motivation, but it weakens the theory-backed framing.


11. In Fig. 1, studying relative error per singular value is not the best target, since what matters is the matrix-level reconstruction. Although Fig. 3 partially fixes this.


12. Fig. 4 “sample ratio” is underspecified. Using a percentage of sequences is a vague measure because it depends on sequence length; an $ l_k/k $ or $ l_k/m $ would be clearer.


13. Looks close to LQ-LoRA + QAT but generalized. Conceptually, the method identifies the important subspace and quantizes the rest, similar to low-rank+quantization approaches like LQ-LoRA but with SVD approximation and STE-QAT.

**Questions:**

1. You mention that Metis adds only modest complexity due to low-rank factors and sampled decomposition, but there is no end-to-end timing. Can you provide actual wall-clock comparisons (forward, backward, and full step) against a plain NVFP4 training baseline on the same hardware/software stack? Given the available kernels.

2. For the quantized GEMMs, how much of the extra time comes from (i) subspace estimation (sampling + projection), (ii) extra GEMM terms from the expanded $ (X_k + X_R)(W_k + W_R) $, and (iii) quantization/dequantization?

3. Can you provide a comparison against strong recent PTQ/few-shot quantization methods (QTIP, Quip#, AQLM, OmniQuant) and to fully-quantized training lines such as rotation-based FP4 (“FP4 All the Way”, HALO-like)

4. For inference, do we still have to run the spectral decomposition / special activation quantization, or can the model be exported to a simpler FP4/PTQ format? How much would the model lose if done so?

---

> ### Author Response · Authors · 2025-11-21
>
> Thank you very much for your time and valuable comments. We provide the relevant figures in the anonymous link: https://anonymous.4open.science/r/ICLR2026_rebuttal-A48C/, referred to as [LINK] below. We summarize our responses as follows:
>
> - **Training Overhead:** We have provided wall-clock and kernel-level measurements on NVIDIA Blackwell Ultra (GB300). Metis introduces 10.12% additional cost over NVFP4.
> - **Benchmark Breadth:** We add results on MMLU, showing gains over FP4 baseline.
> - **Algorithmic Clarity:** We include detailed pseudocode in [LINK].
> - **Additional clarity:** We add validation loss curves, refine definitions (e.g., sample ratio), and clarify the distinctions from LQ-LoRA/QAT.
> - **Inference Behavior:** Although Metis was originally developed to reduce the computational and memory cost of high-precision LLM training, we have added an inference-time variant that adds only 3% FLOPs and outperforms strong PTQ baselines.
> - **Experimental Rigor:** We reran all small-model experiments with multi-seed evaluations, confirming that improvements.
> - **Positioning Among FP4 Methods:** We have compared Metis to FP4 All the Way and HALO, and Metis achieves superior performance.
> - **Hadamard vs. Metis:** We clarify the core limitations of Hadamard on FP4 quantization with quantization-MSE comparisons.
>
> Detailed responses can be found as follows:
>
> **Q1: Runtime evidence on Blackwell GPUs.**
>
> A1: Compared to the NVFP4 baseline, Metis introduces approximately 10.12% computational overhead.
>
> We evaluated the kernel-level, wall-clock computational overheads of Metis on B300 GPU. The extra computational overhead of Metis originates from two operations: (1) spectral decomposition-based quantizations (**4.45%** extra overhead), and (2) the extra small matrix multiplications from spectral decompositions (**5.67%** extra overhead).
>
> **Detailed analysis & overhead breakdown:** We adopted a linear module (shape: [2048, 2048], input shape: [256, 2048, 2048]), which matches the typical shapes in real LLM training, and benchmarked its backward pass time to illustrate the detailed overhead breakdown of Metis. A step of Metis quantization incurs an extra 1.93 ms from the decomposition and its associated quantization and low-rank GEMMs. With the Sparsely Random Sampling strategy, the same low-rank spectral components can be reused across gradient-accumulation steps. Since the dominant subspace estimated from only 1% of the data already matches the full-data subspace with ≈0.9 similarity (Fig.4 in paper), we select a reuse interval of 8 without loss of accuracy. This amortizes the cost to 0.24 ms per step (4.45% overhead vs. NVFP4).
>
> ### **Quantization Strategy Overhead**
> | Operation | Time (ms) | Overhead vs NVFP4 |
> |-|-:|-:|
> |**NVFP4 backward (baseline)** | **5.43** | **0%** |
> | **BF16 backward** | **9.35** | **72.19%** |
> | --- |  |  |
> | **Spectral step components** |  |  |
> | Spectral decomposition (random sampling) | 0.732 | 13.48% |
> | GEMM of U @ S | 0.587 | 10.81% |
> | GEMM of U @ S @ V | 0.253 | 4.66% |
> | Quantization of low-rank branch | 0.359 | 6.61% |
> | Metis quantization | 1.931 | 35.57% |
> | --- |  |  |
> | **Amortized cost (1/8 spectral frequency)** |  |  |
> | **Metis quantization (amortized)** | **0.241** | **4.45%** |
>
> Furthermore, we optimized the small matrix multiplication operations, which led to an overall overhead of 5.67%. We designed a kernel fusion strategy to minimize the extra GMEM traffic introduced by the low-rank branch. This strategy involves: (1) Loading X from GMEM into SMEM only once per tile; (2) Loading corresponding UV and R tiles into SMEM in the same step; (3) Performing computation in SMEM, accumulating results locally, and writing the final output to GMEM once. After fusion, the low-rank branch adds only 5.67% overhead in total. The overhead breakdown is as follows:
> ### **Extra Small Matrix Multiplication Overhead**
> | Operation| Overhead |
> |-|-:|
> | Low-rank parameter reads  | 0.74%|
> | SMEM writes | 0.62% |
> | Computation| 4.30%|
> | Others | 0.01%|
> | **Overall overhead**| **5.67%** |
> The fused kernel code is included in the link https://anonymous.4open.science/r/Metis-quantization-644B.
>
>
> **Q2: Benchmarks are modest.**
>
> A2: We evaluate Metis-pretrained LLaMA-3 8B on the MMLU benchmark. Metis achieves a 0.7% accuracy gain over the FP4 baseline.
> | Method | MMLU |
> |-|-:|
> | BF16 | 30.7 |
> | FP4 | 29.5 |
> | FP4+Metis | 30.2 |
>
> **Q3: Need concrete algorithm.**
>
> A3: We provide pseudocode in [LINK].
>
> **Q4: Need validation loss curves.**
>
> A4: We add validation loss curves for GPT-130M, GPT-1B, and LLaMA-3 8B in [LINK "Validation loss curve of experiments in figure 5"]. Across all models, the validation loss follows the same trend as the training loss shown in the paper.
>
> **Q5: “Sample ratio” is underspecified.**
>
> A5: We redefine the sample ratio as x/y, where x is the number of sampled tokens and y is the total number of tokens in the global batch.

---

> > ### Author Response · Authors · 2025-11-21
> >
> > **Q6: Inference Behavior.**
> >
> > A6: Metis is a training-time method, designed to address the compute and memory bottlenecks of high-precision LLM training (e.g., BF16), and is therefore orthogonal to post-training quantization (PTQ) rather than a competitor to it. The fact that its lightweight inference variant, added only for completeness, still outperforms strong PTQ methods is an additional benefit rather than its primary design goal.
> >
> > At inference time, Metis applies spectral decomposition only to the parameters, adding only 3% FLOPs when using a 1.5% rank.
> > Metis achieves higher inference performance over LQ-LoRA and OmniQuant on LLaMA-3(8B) model.
> >
> > | Method | Loss | ARC-C | ARC-E | BoolQ | LAMBADA | PIQA | RACE | HellaSwag | MMLU | Avg. |
> > |-|-:|-:|-:|-:|-:|-:|-:|-:|-:|-:|
> > | BF16 | 2.44 | 32.4 | 60.4| 59.2 | 37.6 | 70.5 | 47.0 | 50.9 | 30.7 | 48.6 |
> > | LQ-Lora | 2.51 | 32.6 | 60.1 | 58.4 | 34.9 | 71.3 | 45.5 | 48.6 | 29.4 | 47.7 |
> > | OmniQuant | 2.47 | 30.9 | 59.4 | 58.1 | 36.8 | 69.3 | 46.3 | 48.7 | 31.0 | 47.6 |
> > | Metis | 2.45 | 33.2 | 59.2 | 59.0 | 37.3 | 69.9 | 46.9 | 49.5 | 31.3| 48.3|
> >
> > **Q7: Multi-seeded experiments.**
> >
> > A7: We reran the GPT-2 1B experiments with two new seeds trained to 70B tokens, and the training loss ([LINK] "Results of muti-seeded experiments") matches the trends reported in the paper. For the fully trained 100B-token model, we re-evaluated downstream tasks using ten seeds. The results confirm that Metis achieves gains beyond typical variance.
> > | Method | Loss | ARC-C | ARC-E | BoolQ | LAMBADA | PIQA | RACE | HellaSwag | MMLU | Avg. |
> > |-|-|--|-|-|--|-|-|-|-|-|
> > | BF16  | 3.09 | 30.9±0.5 | 59.0±0.5 | 60.3±0.7 | 35.6±0.4 | 65.3±0.4 | 47.5±0.5 | 41.6±0.3 | 25.8±0.4 | 45.8±0.5 |
> > | NVFP4 | 3.22 | 29.8±0.4 | 57.7±0.6 | 59.3±0.5 | 35.2±0.4 | 63.4±0.6 | 47.1±0.6 | 39.7±0.6 | 24.3±0.9 | 44.5±0.6 |
> > | Metis | 3.01 | 30.2±0.5 | 59.3±0.7 | 59.9±0.5 | 35.9±0.5 | 64.1±0.4 | 48.5±0.8 | 41.4±0.5 | 25.5±0.3 | 45.4±0.5 |
> >
> >
> > **Q8: Comparison to FQT methods.**
> >
> > A8: We added comparisons against both ‘FP4 All the Way’ and HALO on GPT-2 1.1B. Since both baselines are trained to 70B tokens, we evaluate Metis at the same 70B-token checkpoint for a fair comparison. Metis consistently shows lower training loss ([LINK] "Results of other FQT methods") and outperforms HALO by 0.6% and FP4 All the Way by 0.9% on downstream tasks.
> > | Method | Loss | ARC-C | ARC-E | BoolQ | LAMBADA | PIQA | RACE | HellaSwag | MMLU | Avg. |
> > |-|-:|-:|-:|-:|-:|-:|-:|-:|-:|-:|
> > | BF16|3.11|28.9±0.5 | 51.6±0.3 | 60.1±0.4|34.7±0.6| 62.9±0.3 | 47.6±0.6 | 38.5±0.3 | 24.9±0.7 | 43.7±0.5 |
> > | FP4 all the way | 3.37 | 26.8±0.3 | 52.8±0.6|58.4±0.5| 34.0±0.5 | 59.4±0.9 | 47.3±0.6 | 37.3±0.6 | 24.6±0.5 | 42.5±0.6 |
> > | HALO | 3.20 | 28.7±0.6 | 50.3±0.6 | 59.4±0.3|33.9±0.5| 60.2±0.6 | 46.7±0.8 | 38.1±0.2 | 25.5±1.0 | 42.8±0.6 |
> > | Metis | 3.08 | 29.4±0.5 | 52.1±0.4 | 60.7±0.7|34.5±0.5| 62.3±0.4 | 47.2±0.5 | 38.3±0.7 | 25.2±0.6 | 43.4±0.5 |
> >
> > **Q9: Hadamard complexity and effectivity.**
> >
> > A9：Hadamard transforms mainly smooth extreme outliers and reduce kurtosis, but the main bottleneck for 4-bit quantization is the very wide dynamic range within each block: when values span far beyond the FP4 range, many small entries are clipped to zero and tail information is lost. Hadamard barely shrinks this overall spread, so widespread clipping remains (Fig. 24 in Appendix). On deep-layer activations from a 1.1B GPT-2 model, the Hadamard recipe yields a quantization MSE of 9.71e-03, whereas Metis reduces it to 3.45e-03. Consistently, NVIDIA’s NVFP4 recipe[1], which heavily relies on random Hadamard, still needs ~16% of deep linear layers in BF16 to avoid instability and shows a ~1.5% loss gap, higher than Metis (all linear layers in FP4 with 0.4% loss gap).
> >
> > Computationally, while fast Hadamard is $O(n\log n)$ for matrix–vector products, work[2] shows that a full RHT mixes along the batch dimension; under data parallelism this would require cross-GPU communication. Practical systems therefore adopt a blockwise RHT: the tensor is split into blocks of size $g$ and each block is processed by a local dense Hadamard matrix multiply, leading to overall overhead $O(n^{2}g)$.
> >
> > **Q10: Matrix-level reconstruction matters.**
> >
> > A10: On deep-layer activation from a 1.1B GPT-2, Metis achieves a lower quantization MSE than Hadamard (3.45e-03 vs. 9.71e-03), indicating more accurate matrix reconstruction.
> >
> > **Q11: Compare with LQ-LoRA + QAT.**
> >
> > A11: LQ-LoRA is a PTQ method; even when combined with QAT, its gradients are still computed in high precision. In contrast, Metis enables true full-FP4 pretraining, and directly tackles the core challenge of gradient quantization by identifying anisotropy as the root cause and addressing it with lightweight spectral decomposition.
> >
> > **Q12: Decomposition bounds are loose.**
> >
> > A12: Very good suggestion. We will definitely consider it in the future work.
> >
> > **References:**
> >
> > [1] Pretraining Large Language Models with NVFP4.
> >
> > [2] Training LLMs with MXFP4.

---

> > > ### Author Response · Authors · 2025-11-28
> > >
> > > We sincerely appreciate your constructive feedback. As the discussion period draws to a close, we would like to follow up to confirm that our responses and the newly added experiments have adequately addressed your concerns. We remain readily available to clarify any remaining questions.
> > >
> > > If you feel our revisions are satisfactory, we would greatly appreciate your consideration in updating your rating. Your evaluation is important to us and has a meaningful impact on the reception of our work. Thank you again for your time and kind suggestions.

---

### Author Response · Authors · 2025-12-03

**We sincerely thank the Area Chair for the time and effort spent evaluating our submission and reviewing both the comments and our responses.**

We confirm that all major concerns about Metis’s practical efficiency (**10.12%** kernel overhead; **5.3%** end-to-end overhead) and fair comparison (**0.6%–0.9%** gains over FP4 baselines) are fully resolved with quantitative evidence.

**Notably, Reviewer eipG (score 2) wrote: “If my concern is addressed, I will consider increasing my score, possibly significantly.” The sole concern was real-world runtime, which our Megatron end-to-end results directly resolve, confirming that Metis is efficient and ready for practical deployment.**

Below, we briefly restate Metis’s core contribution, summarize each reviewer’s key concerns together with our evidence-backed clarifications, and present up-to-date evaluation results of Metis under end-to-end training.

**Contributions:**

Metis is a training-time method designed to address the extremely high computational cost of high-precision LLM training (e.g., BF16), particularly the compute and memory bottlenecks posed by backward gradient computation.
1. **Full 4-bit training.** Metis enables full FP4 training by quantizing weights, activations, and gradients and performing all forward and backward matrix multiplications in FP4, significantly reducing training cost and memory usage.
2. **Root cause of 4-bit degradation.** We identify that FP4 loss primarily arises from extremely wide value distributions, where value-biased quantization over-clips small values and disrupts small singular components.
3. **Superior convergence.** Compared with NVIDIA’s Hadamard-based NVFP4 recipe, which keeps 16% of deep linear layers in BF16 and still has a 1.5% loss gap on an 8B model, Metis uses NVFP4 for all linear layers and reduces the loss gap to 0.4% on the same 8B scale.
4. **Low overhead.** Metis adds only 10.12% overhead relative to vanilla NVFP4.

**Summaries of Reviewer Concerns**

- **Reviewer mj9Y** focused on (1) kernel-level overhead relative to NVFP4 and (2) whether Metis can adapt to post-training quantization (PTQ) scenarios.
- **Reviewer eipG** questioned (1) end-to-end measurements of Metis and (2) whether the assumption $l≫k$ (around line 222) remains valid under gradient-accumulation training.
- **Reviewer kaek** questioned the memory overhead introduced by Metis’ additional operations.
- **Reviewer vMTK** asked for (1) end-to-end efficiency of Metis and (2) comparisons with alternative FP4 or low-bit approaches in both pre-training and PTQ contexts.

**Summaries of Our Responses**

1. **Training Overhead (kernel-level).**
We provided kernel-level benchmarks on NVIDIA Blackwell Ultra (GB300). Metis introduces only 10.12% kernel-level overhead relative to NVFP4.
2. **Inference Behavior, PTQ Relation, and Positioning among FP4/Low-Bit Frameworks.**
Metis is a Full-Quantization Training method and is orthogonal to PTQ approaches. Nevertheless, we also provide an inference-time variant that requires only 3% additional FLOPs while outperforming strong PTQ baselines by 0.6%–0.7%. In the training regime, Metis surpasses other FP4 training methods by 0.6%–0.9% on downstream tasks.
3. **Assumption $l≫k$ under gradient accumulation.**
We confirm that this assumption always holds in LLM training. For example, in Llama-8B, $l=128×8192≫k=4096$, where $l$ is the global batch size times the sequence length. Gradient accumulation does not change $l$, since Metis reuses the same low-rank spectral components across accumulation steps rather than recomputing the decomposition each step.
4. **Memory overhead of additional Metis operations.**
Although Metis introduces additional steps, our kernel-fusion approach effectively eliminates redundant memory traffic, resulting in only 1.36% memory overhead and 5.67% total overhead.

**New result: Megatron end-to-end implementation.**

We additionally integrated Metis for both activations and gradients within the Megatron training framework and conducted experiments on a DGX SPARK equipped with GB10, which support FP4 but have different performance characteristics compared to GB300. To fit within single-GPU memory constraints while keeping all other hyperparameters unchanged, we reduced the Llama-3-8B depth to 12 layers and performed full end-to-end overhead measurements. The results show that Metis incurs only **5.3%** additional end-to-end computation cost and **1.2%** memory cost relative to plain NVFP4, demonstrating that Metis remains highly efficient even in full training pipelines.

**This result directly addresses the questions raised by Reviewer eipG and Reviewer mj9Y regarding end-to-end efficiency and confirms that Metis is efficient and ready for practical deployment.**
|Method|Elapsed time per iteration (ms) | Throughput per GPU (TFLOP/s/GPU) | Max allocated memory (MB) |
|-|-|-|-|
|NVFP4|17672|43.3| 43263|
|Metis|18608|41.1|43791|
|**Relative overhead**| **+5.3%**|**-5.1%**|**+1.2%**|

---

### Meta-Review · Area_Chair_pNGS · 2026-01-19

**Summary:**

This paper receives scores of 6,6,4,2, while the most critical concern is about the end-to-end runtime (all reviewers point out this concern), which is well-addressed with quantitative results in the author responses. The extra overhead (10.2% compared to naive NVFP4) is not small, but it achieves gains in accuracy of 0.6%–0.9%, which is close to BF16. The remaining concerns are not critical to the main contribution of this paper, and I suggest accepting the paper.

**Reviewer Concerns:**

The main concerns about real computational cost are well-addressed. Most of the other concerns are also addressed.
There are remaining concerns,
(1) NVFP4 training achieves lower loss compared to BF16 training in GPT models. This is a strange result, and the author's explanation in the response is a rough guess. A more in-depth analysis is critical to make the overall experiment convincing.
(2)  Only one stronger discriminative benchmark, MMLU, is included. The author evades the mentioned GSM8K. In the final revision,  I suggest including more benchmarks, such as GSM8K, GPQA, and IFEval.
(3) Validation on larger models.

**Reviewer Scores:**

The main concerns about end-to-end runtime are well-addressed and Reviewer mj9Y and eipG may raise their ratings.

---

### Decision · Program_Chairs · 2026-01-26

Accept (Poster)